# Cytotoxic and Antibacterial Prenylated Acylphloroglucinols from *Hypericum olympicum* L.

**DOI:** 10.3390/plants12071500

**Published:** 2023-03-29

**Authors:** Yana Ilieva, Georgi Momekov, Maya Margaritova Zaharieva, Teodor Marinov, Zlatina Kokanova-Nedialkova, Hristo Najdenski, Paraskev T. Nedialkov

**Affiliations:** 1Department of Infectious Microbiology, The Stephan Angeloff Institute of Microbiology, Bulgarian Academy of Sciences, 1113 Sofia, Bulgaria; illievayana@gmail.com (Y.I.); zaharieva26@gmail.com (M.M.Z.);; 2Department of Pharmacology, Toxicology and Pharmacotherapy, Faculty of Pharmacy, Medical University of Sofia, 1000 Sofia, Bulgaria; gmomekov@pharmfac.mu-sofia.bg; 3Pharmacognosy Department, Faculty of Pharmacy, Medical University of Sofia, 1000 Sofia, Bulgaria

**Keywords:** *Hypericum olympicum*, polyprenylated acylphloroglucinols, cytotoxic, antibacterial, MRSA, anti-biofilm, drug-likeness

## Abstract

Two new bicyclo[3.3.1]nonane type bicyclic polyprenylated acylphloroglucinol derivatives (BPAPs), olympiforin A and B as well as three known prenylated phloroglucinols, were isolated from the aerial parts of *Hypericum olympicum* L. The structures of the isolated compounds were established by means of spectral techniques (HRESIMS and 1D and 2D NMR). All compounds were tested on a panel of human tumor (MDA-MB-231, EJ, K-562, HL-60 and HL-60/DOX) and non- tumorigenic (HEK-293 and EA.hy926) cell lines using the MTT assay. All tested compounds exerted significant in vitro cytotoxicity with IC_50_ values ranging from 1.2 to 24.9 μM and from 0.9 to 34 μM on tumor and non-cancerous cell lines, respectively. Most of the compounds had good selectivity and were more cytotoxic to the tumor cell lines than to the normal ones. A degradation of the precursor caspase 9 for some of the compounds was observed; therefore, the intrinsic pathway of apoptosis is the most likely mechanism of cytotoxic activity. The BPAPs were examined for antibacterial and antibiofilm activity through the broth microdilution method and the protocol of Stepanović. They showed a moderate effect against *Enterococcus faecalis* and *Streptococcus pyogenes* but a very profound activity against *Staphylococcus aureus* with minimum inhibitory concentrations (MIC) in the range of 0.78–2 mg/L. Olympiforin B also had a great effect against methicillin-resistant *S. aureus* (MRSA) with an MIC value of 1 mg/L and a very significant antibiofilm activity on that strain with a minimum biofilm inhibition concentration (MBIC) value of 0.5 mg/L. The structures of the isolated compounds were in silico evaluated using ADME and drug likeness tests.

## 1. Introduction

*Hypericum* L. (Hypericaceae) is a genus of ca. 480 species distributed in all temperate parts worldwide, and plants have been used for centuries in traditional medicine as a healing agents for wounds, burns and ulcers [1]. The most known amongst the species from this genus is *H. perforatum* (St. John’s Wort), the only herbal alternative to the classic synthetic antidepressants [2]. Important secondary metabolites from this genus are acylphloroglucinols, especially polycyclic polyprenylated acylphloroglucinols (PPAPs). They possess interesting structures with complex, bridged, highly oxidized and substituted systems [3]. About 450 PPAPs have been isolated so far almost exclusively from Hypericaceae and Clusiaceae [4]. Major classes of PPAPs are bicyclo[3.3.1]nonane-type bicyclic polyprenylated acylphloroglucinols (BPAPs) and the adamantane type. The pioneer hyperforin was the first BPAP with revealed profound antibiotic [5,6,7] and cytotoxic [8,9,10] activities. Later on, numerous simple or complex acylphloroglucinols from *Hypericum* spp. with outstanding cytotoxic and antineoplastic [11,12,13,14,15,16] and potent antibacterial [17,18,19,20,21,22,23] properties were discovered, as outlined in several more or less recent reviews [2,4,16,24,25,26,27].

The combination of antineoplastic and antibacterial activity is very relevant in the present time since cancer is the second leading cause of death globally after heart disease, [28] and antimicrobial resistance (AMR), found in many species of bacteria, is an increasingly serious threat to human health. In addition, bacterial biofilms, which contribute to >80% of all infections and 65% of infections in developed countries [29,30], are another challenge for which some phloroglucinol derivatives [31,32] may provide relief. Other pharmacological effects of phloroglucinols include antioxidant, antidepressant, antiviral (including effectiveness against SARS-CoV-19), antinociceptive etc. [23,27,33,34,35].

*H. olympicum* L. [section Olympia (Spach) Nyman] is a small perennial herb with areal the Balkan peninsula and northwestern Turkey. Aerial parts of the plant are used in Turkish folk medicine for stomachache, inflamed wounds and cuts [36]. In fact, the first study of the *Hypericum* genus for antibacterial properties of belongs to aqueous extract of this species (together with *H. elodes* (Spach) W. Koch) [37]. Previous studies have demonstrated the presence of small amounts of hyperforin [38] and simple acylphloroglucinols among which olympicin A is an example of a phloroglucinol with a profound antibacterial activity [39].

In the course of our scientific search for cytotoxic and antimicrobial acylphloroglucinols from Bulgarian *Hypericum* species, we present the results from the phytochemical and pharmacological study of *H. olympicum* L. Five compounds **1**–**5** were isolated, and their structures (Figure 1) were elucidated by means of spectral analyses. All compounds were studied for cytotoxic activity while for the antibacterial effect only the compounds that had not been tested before were examined. The structures of the isolated compounds were evaluated in silico using the web tool SwissADME.

## 2. Results

### 2.1. Identification of the Isolated Compounds

The air-dried aerial parts of *Hypericum olympicum* L. were successfully extracted with the CH_2_Cl_2_-MeOH (4:1) mixture and the resulting extract was subjected to extensive chromatographic procedures that led to the isolation of three known (**1**–**3**) and two new (**4** and **5**) prenylated acylphoroglucinols (Figure 1). The known compounds were identified as olympicin A (**2**) [39] and hyperpolyphyllirin/hyperibine J (**3**) [40].

Compound **1** was obtained as a colorless oil showing in its HRESIMS spectrum a deprotonated molecule [M−H]^−^ at *m*/*z* 331.1913 pointing out a molecular formula C_20_H_28_O_4_ with seven degrees of unsaturation. The MS/MS of the deprotonated ion of **1** showed a base fragment at *m*/*z* 194.058 with the elemental composition of C_10_H_10_O_4_, which corresponded to the cleavage of a geranyl side chain (neutral loss of 137 Da). The sequential neutral loss of 42 Da (*m*/*z* 194.058 → *m*/*z* 152.011) was indicative of the cleavage of the propyl side chain. The ^1^H-NMR spectrum (Table 1) of compound **1** showed two doublets at *δ*_H_ 5.93 and 6.00 with the *meta*-coupling constant *J* = 2.3 Hz, which were attributed to the hydrogens of the phloroglucinol ring. These two signals correlated in the HSQC experiment with carbon resonances at *δ*_C_ 91.7 and 96.5, respectively. The ^13^C NMR spectrum of **1** (Table 1) showed signals of three oxygenated carbons that appeared downfield at *δ*_C_ 162.55, 162.5 and 167.4 and were attributed to C-2, C-4 and C-6, respectively. The resonance of the quaternary aromatic carbon connected to the acyl side-chain appeared at *δ*_C_ 105.2. All these signal were typical for an intact acylphloroglucinol ring [41]. In addition, the ^1^H-NMR spectrum showed the presence of the three-proton singlets of methyl groups at *δ*_H_ 1.62 (H_3_-8″), 1.69 (H_3_-9″) and 1.74 (H_3_-10″), two methylene multiplets at *δ*_H_ 2.10 (H_2_-4″) and 2.13 (H_2_-5″), a doublet of oxygen-bearing methylene at *δ*_H_ 4.56 (H_2_-1″), as well as two multiplets of olefinic protons at *δ*_H_ 5.10 (H-6″) and 5.50 (H-2″), all belonging to the geranyloxy moiety. In the ^13^C-NMR spectrum, the carbon signals of the geranyloxy side-chain appeared as three methyl groups at *δ*_C_ 16.6 (C-10″), 17.7 (C-8″) and 25.7 (C-9″); two methylenes at *δ*_C_ 26.3 (C-5″) and 39.4 (C-4″); an oxygen-bearing methylene at *δ*_C_ 65.6 (C-1″); two olefinic carbons at *δ*_C_ 118.1 (C-2″) and 123.6 (C-6″); as well as two quaternary carbons at *δ*_C_ 132.0 (C-7″), and 142.3 (C-3″). The cross-peaks between protons of an oxygen-bearing methylene at (H_2_-1″) and the proton at position 3 in the NOESY experiment as well as the correlation with a quaternary carbon at position 2 in the HMBC experiment (Figure 2) unambiguously confirmed that the geranyloxy chain was placed at position 2. The signals in the ^1^H-NMR (a six-proton doublet at *δ*_H_ 1.15 (H_3_-3′ and H_3_-4′) and a septet at *δ*_H_ 3.80 (H-2′)) as well as those in the ^13^C-NMR spectrum (methyl groups at *δ*_C_ 19.3 (C-3′ and C-4′), a methine at *δ*_C_ 39.5 (C-2′) and a keto group at *δ*_C_ 210.6 (C-1′)) were typical for 2-methylpropanoyl group. The remaining signal in the ^1^H-NMR spectrum at *δ*_H_ 14.14 was assign to a chelated hydroxyl group at C-6. This observation was supported by a bathochromic shift of the maximum at 290 nm in the UV spectrum of **1** after addition of AlCl_3_/HCl (Δ_λ_ = 72 nm). A bathochromic shift of the maximum at 290 nm was observed after addition of NaOAc (Δ_λ_ = 33 nm), which is indicative of a free hydroxyl group in the *para* position to the carbonyl function. Thus, compound **1** was identified as (*E*)-1-[2-{(3,7-dimethylocta-2,6-dien-1-yl)oxy}-4,6-dihydroxyphenyl]-2-methylpropan-1-one. This is a known natural product, which has been firstly isolated from *Hypericum densiflorum* Pursh [42] but has was not described, so far. We present its NMR and other spectral data for the first time.

Compound **4** was isolated as a colorless oil. In the HRESIMS spectrum of **4**, the signal of the monoisotopic protonated molecule [M+H]^+^ appeared at *m/z* 497.3622 (calculated for C_32_H_49_O_4_, 497.3625) pointing out C_32_H_48_O_4_ as its molecular formula, which suggested nine degrees of unsaturation. The MS/MS of the protonated molecule showed a fragment ion at *m/z* 441.3000 (C_28_H_41_O_4_), which corresponded to a loss of four terminal carbons of a prenyl group, a typical pattern of cleavage for PPAPs or just prenylated acylphloroglucinols [43,44,45]. The fragments at *m/z* 373.237, 293.175 and 237.112 corresponded to a sequential loss of two and three prenyl groups as well as a propane from the 2-methylbutanoyl side chain, respectively. The fragments at *m/z* 163.148 and 69.071 were due to a trioxygenated bicyclo[3.3.1]nonane skeleton and a free prenyl group, respectively.

Due to of keto-enol tautomerism and instability, that were common for phloroglucinol derivatives with free keto and hydroxyl groups, especially hyperforin-type BPAPs [40,46,47], the ^13^C- and ^1^H-NMR spectra of **4** (Table 2 and Table 3) were not clear. In order to acquire more stable derivatives, **4** was treated with a diazomethane (Figure 3) and the resulting mixture (ratio 1:4) was separated by SP-HPLC and gave methoxy derivatives **4a** and **4b**.

The ^13^C-NMR spectrum of **4a** (Table 2) had signals for 33 carbons, including 11 quaternary (including 3 carbonyl, 4 alkene and 1 oxygenated sp^2^ carbon atom), 6 methine, 5 methylene and 11 methyl (including one oxygenated) groups. The resonances at *δ*_C_ 207.8 and 209.0 due to the quaternary carbons C-9 and C-10 indicated unconjugated carbonyls. The signals at δ_C_ 194.3, 128.0 and 173.2 belonging to the quaternary carbons C-2, C-3 and C-4, respectively, suggested a 1,3-keto-enol system. In addition, the carbon spectrum showed signals at *δ*_C_ 84.0, 54.5 and 48.6, which corresponded to the other three quaternary carbons C-1, C-5 and C-8, respectively, as well as signals of the methine carbon C-7 at *δ*_C_ 43.4 and the methylene C-6 at *δ*_C_ 40.6. Altogether these data designated an acylphloroglucinol with a bicyclo[3.3.1]nonane skeleton (hyperforin type). It should be noted that the numbering system used for hyperforin-type compounds in this work follows that of the recent reviews about PAPPs [4,26].

The ^1^H-NMR spectrum of **4a** (Table 3) exhibited one-proton multiplets of three methines at δ_H_ 5.01 (H-16), 4.95 (H-22) and 5.05 (H-29) belonging to prenyl groups. In addition, the proton spectrum showed signals of a methine at *δ*_H_ 1.74 (m, H-11), a methylene at δ_H_ 1.76 m and 1.25 m (m, H_2_-12), and two methyl groups at *δ*_H_ 0.78 (t, *J* = 7.4 Hz, H_3_-13) and 1.12 (d, J = 6.5 Hz, H_3_-12); all were indicative of a 2-methylbutanoyl side chain. The double doublets at δ_H_ 3.21 (dd, *J* = 15.0, 6.9 Hz H-15_a_) and 3.11 (dd, *J* = 15.0, 6.3 Hz, H-15_b_) gave cross-peaks with the signal of C-3 in the HMBC experiment (Figure 4)—an indication that the first prenyl group was attached to position 3.

The resonances of methylene protons H_2_-21 at δ_H_ 2.12 and 1.75 1 gave cross-peaks in the HMBC experiment (Figure 4) with the signal at δ_C_ 43.4 (C-7), which was unambiguous evidence that the second prenyl group was placed at position 7. The signals of the methylene protons H_2_-28 at δ_H_ 2.12 and 1.80 gave ^1^H-^1^H COSY correlations (Figure 4) with methylene protons H_2_-27 at δ_H_ 1.96 and 1.32. The later proton signals gave clear HMBC correlation with the carbon resonance at δ_C_ 48.6 (C-8). This was evidence that the third prenyl was attached through C-27 at position 8. The signal of methyl protons H_3_-26 at δ_H_ 1.01, 3H, s, H-26 exhibited a HMBC cross-peak with the signal of C-8, which suggested that it was directly connected to a bicyclo[3.3.1]nonane. The three-proton singlet signal of H_3_-20 at δ_H_ 1.30 showed HMBC correlations with the resonances of the carbonyl C-9 at δ_C_ 207.8, the methylene carbon C-6 at δ_C_ 40.4 and also with the quaternary carbon C-5 at δ_C_ 54.5. This evidence indicated that compound **4a** belonged to a large class of PPAPs, where instead of prenyl group at C-5 these have a methyl function [4,26]. The methoxy group formed after the methylation appeared as a three-proton singlet at δ_H_ 3.92, which had a cross-peak with the signal of C-2. The 2-methylbutanoyl group had to be attached to C-1, which made the compound a PPAP of type A according to the classification in the review of Ciochina and Grossman [26].

Concerning the relative configuration of **4a**, the chemical shift difference Δδ_H_ 0.57 of H_ax_-6 and H_eq_-6, together with the chemical shift of C-7 at δ_C_ 43.4, showed that the prenyl group at position 7 was *exo* orientated according to the rule postulated by Grossman and Jacobs [26]. This meant that the more saturated ring of the bicyclic compound had a chair conformation rather than a boat and was also confirmed by NOESY correlations (Figure 5) of the signals of methyl protons H_3_-26 with those of the methylene protons H_2_-21 and H_ax_-6. The aforementioned correlation in the NOESY spectrum showed that this methyl group was axially orientated. Thus, the structure of the derivative **4a** was established as (1*R*,5*R*,7*S*,8*R*)-4-methoxy-5,8-dimethyl-3,7-bis(3-methylbut-2-en-1-yl)-1-(2-methylbutanoyl)-8-(4-methylpent-3-en-1-yl)bicyclo[3.3.1]non-3-ene-2,9-dione.

Compound **4b** had ^13^C- and ^1^H-NMR spectra (Table 2 and Table 3) similar to that of **4a**. The signal of methoxyl protons that appeared at δ_H_ 4.03 in the ^1^H-NMR spectrum gave a HMBC correlation (Figure 4) with δ_C_ 170.2 at C-2, while the methyl group H_3_-20 gave a cross-peak with the resonance of the carbonyl C-4 at δ_C_ 196.6. The only difference between **4a** and **4b** was the position of the methoxy group, which indicated the existence of two tautomeric forms of compound **4**. Thus, the structure of the derivative **4b** was established as (1*R*,5*S*,6*R*,7*S*)-4-methoxy-1,6-dimethyl-3,7-bis(3-methylbut-2-en-1-yl)-5-(2-methylbutanoyl)-6-(4-methylpent-3-en-1-yl)bicyclo[3.3.1]non-3-ene-2,9-dione.

Thus, compound **4** was established to be mixture of keto-enol tautomers with structures (1*R*,5*R*,7*S*,8*R*)-4-hydroxy-5,8-dimethyl-3,7-bis(3-methylbut-2-en-1-yl)-1-(2-methylbutanoyl)-8-(4-methylpent-3-en-1-yl)bicyclo[3.3.1]non-3-ene-2,9-dione and (1*R*,5*S*,6*R*,7*S*)-4-hydroxy-1,6-dimethyl-3,7-bis(3-methylbut-2-en-1-yl)-5-(2-methylbutanoyl)-6-(4-methylpent-3-en-1-yl)bicyclo[3.3.1]non-3-ene-2,9-dione. It is a new natural product and was given the trivial name olympiforin A.

Compound **5** was isolated as a colorless amorphous mass. Its HRESIMS spectrum showed a protonated molecule [M+H]^+^ at *m/z* 469.3308 (calculated for C_30_H_45_O_4_, 469.3312) pointing out C_30_H_44_O_4_ as the molecular formula and nine degrees of unsaturation. The MS/MS spectrum of the protonated molecule [M+H]^+^ showed fragment ions at *m*/*z* 345.206 (C_26_H_37_O_4_), 413.268 (C_21_H_29_O_4_) and 277.143 (C_16_H_21_O_4_), which was indicative for sequential neutral loss of one, two and three prenyl units, respectively.

The ^13^C-NMR spectrum of **5** (Table 4) showed signals for 30 carbons: 10 quaternary, 6 methine, 5 methylene and 9 methyl. Its ^1^H-NMR (Table 4) and ^13^C-NMR spectra indicted a hyperforin type of BPAP and were very similar to those of **4** and its methoxy derivatives **4a** and **4b**.

The similar features of the bicyclo[3.3.1]nonane acylphloroglucinol included two nonconjugated carbonyl groups at δ_C_ 206.2 (C-9) and 209.5 (C-10), an enolized 1,3-dicarbonyl system (δ_C_ 169.2 (C-2), 117.9 (C-3) and 192.2 (C-4)), two quaternary carbons at δ_C_ 83.7 (C-1) and 48.6 (C-8), a methine carbon at δ_C_ 42.5 (C-7), a methylene group at δ_C_ 36.5 (C-6) and three prenyl groups with the same as in compound **4** positions (methines at δ_H_ 5.25 (H-15), 5.06 (H-20), and 4.96 (H-27)). In the same way, there was a methyl function at position 8 (δ_H_ 1.05 (H_3_-24) and δ_C_ 13.6 (C-24)) forming an prenylmethyl group with the third prenyl chain at position 25. The patterns of the COSY and HMBC correlations (Figure 6) were very similar to those of **4a**. The chemical shift difference in resonances of H_ax_-6 and H_eq_-6 (Δδ_H_ = 0.60), together with the chemical shift of C-7 (δ_C_ 42.5), showed that the prenyl group at C-7 was in the *exo* orientation. Instead of a methyl group placed at position 5, there was a multiplet of a proton at δ_H_ 3.24, which gave HMBC cross-peaks with carbon resonances at δ_C_ 117.9 (C-3), 169.2 (C-4) and 206.2 (C-9).

Furthermore, **5** had signals for a 2-methylpropanoyl side chain attached to C-1 (a septet at δ_H_ 2.08 (*J* = 6.5 Hz, H-11) as well as two doublets at δ_H_ 1.11 (*J* = 6.5 Hz, H_3_-12) and 1.03 (*J* = 6.5 Hz, H_3_-13)). Thus, the compound **5** was identified as (1*S*,5*R*,7*S*,8*R*)-4-hydroxy-1-isobutyryl-8-methyl-3,7-bis(3-methylbut-2-en-1-yl)-8-(4-methylpent-3-en-1-yl)bicyclo[3.3.1]non-3-ene-2,9-dione. The metabolite is a new natural compound that was given the trivial name olympiforin B. It can also be presented as 5-deprenyl-hyperforin as it is in the homologous list of deprenyl hyperforin derivatives, which also includes hyperfirin (7-deprenyl-hyperforin) [48], secohyperforin (30-deprenyl-hyperforin) [49] and hyperevolutin A (3-deprenyl-hyperforin) [50]. In addition, olympiforin B has the structure of compound **3** but without methyl group at position 5. Although the NMR spectra presented only one tautomer of 1,3-keto-enol system, it cannot be excluded that olympiforin B also exists in two tautomeric forms as hyperforin itself or other hyperforin-type PPAPs with free keto and hydroxyl groups.

### 2.2. Cytotoxic Activity of the Isolated Compounds

All tested compounds had in vitro cytotoxic activity at low micromolar ranges—IC_50_ varied in the limits of 1.11–24.94 μM for the different cell lines with an origin from solid tumors or hemoblastoses (Table 5). Olympiforin B (**5**) possessed the most pronounced effect on the tumor cell lines in the present study (IC_50_ 1.16–1.71 μM).

EA.hy926 is a hybrid cell line between endothelial cells and lung cancer cells, but it is not tumorigenic [51]. In regard to the selectivity towards tumor cell lines in comparison to non-tumorigenic cells, the results were quite varied, but most of the substances exerted a more specific inhibitory effect on the cancer cell lines (Table 5 and Table 6). All compounds, except **5**, had an SI over 1 in relation to HEK-293 and all compounds, except **1** and **4,** had an SI over 1 in relation to EA.hy926. A selectivity index (SI) over 2–3 is considered promising, and compounds **2**–**5** achieved those values and even higher, especially relative to EA.hy926. Compound **5** had the best SI (as high as 7.14) towards EA.hy926 cells. Additionally, clinical drugs, such as cisplatin, often have a low SI (sometimes <1), i.e., they are more cytotoxic to a given normal cell line than to certain tumor cell lines [52,53,54], meaning that the compounds with SI < 2 still could be drug candidates.

Given the fact that natural products, including some of the most important antineoplastic agents, are substrates of the various efflux ATP-transporters, we aimed to study the chemosensitive profile of the multi-drug resistant (MDR) cell line HL-60/DOX, which is characterized by the excessive expression of MRP-1. As reflected in the resistance indices (RI), it turned out that HL-60/DOX had collateral sensitivity for all tested compounds except **5,** i.e., this resistant cell line was more sensitive to them than the parent cell line HL-60. Hyperpolyphyllirin/hyperibine J (**3**) excelled as its activity increased around tenfold for the resistant cell line (Table 5 and Table 6).

The growth of solid tumors is dependent on the process of neovascularization or angiogenesis, i.e., forming of autonomous system of blood vessels. This process is determined by secretion of proangiogenic signal molecules by tumor cells and relies on three fundamental phenomena concerning the vascular epithelium—migration, proliferation and formation of capillaries.

Suppression of vascular endothelium is a key mechanism of antiangiogenic activity. That is why we followed the inhibitory potential of the tested phloroglucinols on the cell line EA.hy926, which is an established immortalized model of vascular endothelium. The conducted research showed that the tested acylphloroglucinols inhibit the proliferation of the endothelial cells at micromolar concentrations. Compounds **4** and **1** had the greatest relative activity with values of IC_50_ 5.34 μM and 6.73 μM, respectively. The other com-pounds demonstrated a less pronounced inhibitory effect but also exerted significant sup-pression of the cell proliferation in an MICromolar range (Table 5).

### 2.3. Antiproliferative Activity of the Isolated Compounds

It can be observed that the antiproliferative activity of the isolated compounds generally showed the same pattern as their cytotoxic effect, with a few exceptions (Table 7). Compounds **1** and **2** have already been tested for antiproliferative activity [42], and in the present study they generally demonstrated a stronger effect.

### 2.4. Influence of the Isolated Compounds on the Apoptosis-Related Proteins Procaspase 9 and Bcl-2

We observed degradation of the precursor caspase 9 for **2**–**4** (for only one cell line for the last compound) and of the antiapoptotic protein Bcl-2 for **2** as shown in (Figure 7). This fact suggests that the aforementioned compounds induce programmed cell death as their mechanism of cytotoxic activity. As caspase 9 is connected to the intrinsic signal pathway of apoptosis, the precise mechanism has to be the recruitment of the mitochondrial signal pathway. There was not any observed decrease in the level of procaspase 9 for **1** and **5**.

The apoptogenic effects of the three compounds are different and so are the effects of a single compound when treating the separate tumor test systems. Compound **2** at cell line MDA-MB-231 exerts the most pronounced effect for caspase 9 as well as for Bcl-2 (the largest decrease of relative density of the bands).

Two types of results are observed. In the first type, the proapoptotic effect is in straight correlation with the concentration of treating. That is the case for procaspase 9, influenced by compound **3** in both cell lines and for procaspase 9 and Bcl-2 influenced by compound **2** in HL60/DOX. In fact, the lower concentration (1/2 IC_50_) did not change the level of procaspase 9 in the latter probe. In the rest of the cases, the apoptogenic effect is greater at the lower concentration (1/2 IC_50_), which is subtoxic but possibly activates the apoptotic pathways. It is likely that the concentration which causes death at 50% of the cells treated (IC_50_) causes necrotic death cells to a greater extent and consequently there was no change in the level of apoptosis-associated precursor caspases. In the sample of MDA-MB-231 treated with compound **4** at the concentration equal to IC_50_, there was no difference in the quantity of procaspase 9.

### 2.5. Antibacterial Activity

Compounds **1** and **2** have already been tested for their antibacterial effect [31,39,42,55]. Therefore, compounds **3**–**5** were studied for their bacteriostatic properties (Table 8). Minimal inhibitory concentrations (MICs) were in the range of 0.78–31 mg/L (1.57 to 64.22 μM). The best activity was observed against *Staphylococcus aureus*, where the lowest MIC observed was 0.78 mg/L or 1.57 μM for **4** and a range of MIC values of 0.78–2 mg/L (1.57–4.14 μM). *S aureus* once again turned out to be the most susceptible strain as in our previous research [56].

In regard to MRSA, compound **5** also had a strong activity with an MIC value of 1 mg/L (2.14 μM). The compounds were not active against the Gram-negative bacterium (E. coli) at concentrations up to 100 mg/L. The metabolic activity of the influenced tested strains is presented in Figure 8.

The potential to inhibit bacterial biofilm production of the same substances was assessed (Table 9 and Figure 9). The range of their minimum biofilm inhibition concentrations (MBICs) was 0.5–12.5 mg/L, and their median biofilm inhibitory concentrations (MBIC_50_) were in the range of 0.05–0.67 mg/L.

### 2.6. In Silico ADME and Drug-Likeness Evaluation

The results from the virtual in silico screening are presented in Table 10, Table 11, Table 12, Table 13 and Table 14 and in Figure 10.

The calculated high consensus LogP values showed that all the tested compounds were lipophilic. Only compounds **1** and **2** have values below 5, which is demanded by most filters in the pharmaceutical industry. Compounds **3**–**5** have LogP values above 6 (Table 11). As expected, the LogS estimation model assigned the compounds to be from insoluble to poorly and moderately soluble, depending both on the model and the tested compound (Table 12).

The data calculated in silico suggested that **1** and **2** had high gastrointestinal absorption and, as we have already emphasized, this is an advantage as the oral route of administration of drugs is preferred [57]. The rest of the compounds are expected to have poor gastrointestinal absorption.

Compound **1** is the only substance that should be able to cross the blood–brain barrier (BBB), which is a favorable feature for the treatment of brain tumors and metastases, but this also implies risks of CNS side effects. In addition, **1** is the only one of the isolated compounds assumed to not be a substrate for the P-gp transporter. Therefore, **1** would have the advantage to not be exported from the gastrointestinal lumen, CNS and from tumor cells (over)expressing this transporter.

This study is another example where all the compounds are expected to inhibit CYP3A4, the most important isoform of the cytochrome P_450_ family. Since compounds **3**–**5** are of hyperforin type, we expect the inhibition to occur shortly after administration, while we expect induction of expression to be detected after chronic treatment as this is the behavior of hyperforin [58,59,60,61]. CYP2C19 and CYP2D6 are calculated to be inhibited, and for all the other isoforms the results vary depending on the compound. Again, the negative LogKp values of the compounds indicated very low skin permeation (Table 13).

We report another example of acylphloroglucinols from Hypericum, which obey the Lipinski filter, which was already emphasized as the pioneer and archetype of all drug-likeness filters [57]. The compounds were also successful in the Veber filter but not in the one of Muegge. Only **1** and **2** complied to the Ghose and Egan filters. The Abbot bioavailability score was 55% for **1** and **2** and 56% for **3**–**5**. 

The results from the in silico medicinal chemistry tests showed that **1**–**5** passed the PAINS filter but did not comply to the Brenk filter. They also did not satisfy the Lead-likeness filter because of their lipophilicity, chemical complexity and, for **3**–**5**, their high molecular mass also. As the phloroglucinols from *Hypericum annulatum* Moris, those in the current study need decreased lipophilicity and the elimination of problematic functionalities (and structure simplification for **3**–**5**) if at some point they are taken as lead scaffolds for semi-synthesis. Their synthetic accessibility coefficient suggested that **1** and **2** should be relatively easy to be synthesized in a laboratory while **3**, **5** and especially **4** should be relatively difficult (Table 14).

The SwissADME images of the bioavailability radars of the isolated compounds are presented in Figure 10. In order to be estimated as drug-like, the red line of a compound must be fully included in the pink area. Any deviation represents a suboptimal physicochemical property for oral bioavailability. The pink area comprises the optimal range of the following properties: lipophilicity—XLOGP3 between –0.7 and 5.0; size—MW between 150 and 500 g/mol; polarity—TPSA between 20 and 130 Å2; solubility—log S should not exceed 6; saturation—fraction of sp^3^ hybridized carbons should be at least 0.25; and flexibility—not more than nine rotatable bonds.

## 3. Discussion

Interestingly, besides wounds [2,62,63], the *Hypericum* species has been used in traditional medicine in some places to treat cancer and tumors as well as, for example, some Cameroonian *Hypericum* species and *H. sampsonii* Hance and *H. scabrum* L. in Asia [54,64,65]. This demonstrates a connection between the ethnopharmacological use of the species and its subsequently elucidated phytochemical composition. The known use of *H. olympicum* in Turkey for inflamed wounds contributes to perusing its microbiological study. 

Previous research shows that PPAPs (including from plant taxons other than *Hypericum*) have IC_50_ values towards tumor cell lines often in the range of 1–10 μM and even < 1.5 nM for some PPAPs from *Garcinia* (Clusiaceae) [4].

BPAPs or prenylated acylphloroglucinols from the species *H. pseudopetiolatum* Keller, *H. ascyron* L., *H. sampsonii*, *H. yojiroanum* Tatew. ex Koji Ito, *H. drummondii* (Grev. ex Hook.) Torr. ex A. Gray, *H. polyanthemum* Klotzsch ex H.Reich., *H. attenuatum* Fischer ex Choisy, *H. papuanum* Ridl., *H. amblyocalyx* Coustur. ex Gand. etc. have a broad range of IC_50_ values (0.14–227 μM) on human tumor cell lines NCI-H460 and A-549 (lung), AGS (stomach), HCT-116 HT-29 and Caco-2 (colon), U-373 MG (glioblastoma), HepG2 (liver), Eca109 (esophagus), HeLa (cervix), Jurkat (T-cell leukemia), KB (epidermoid carcinoma), etc., and on murine cell lines, such as L1210 and P388 (lymphoma) [2,24]. In some cases, e.g., on MDR cancer cell lines (KB-C2 and K562/Adr), the effect was more potent than the positive control doxorubicin [2]. The lowest IC_50_ value of hyperforin is 1.8 μM against leukemia U937 cells [66] but the IC_50_, for example, of the prostate cancer cell PC-3 is 37 μM [67].

Recent research also confirms various IC_50_ values in some cases as low as 1.7−10 μM and in other cases 20−40 or > 160 μM for BPAPs, seco-spirocyclic PPAP derivatives or just prenylated acylphloroglucinols from *H. sampsonii*, *H. faberi* R. Keller, *H. perforatum*, *H. choisianum* Wall. ex N. Robson, *H. longistylum* Oliv. and *H. petiolulatum* Hook.f. ex Thomson ex Dyer against HL-60, SW-480 or other human tumor cell lines, such as SMMC-7721 (hepatic), A-549 (lung), MCF-7 (breast), PANC-1 (pancreatic) and NB4 (leukemia) [11,12,13,14,16,17,68]. Again, in some cases the IC_50_ values were equal to that of the control (e.g., cisplatin) and there was selectivity, i.e., weaker cytotoxicity on a normal cell line, e.g., Beas-2B (bronchial epithelium) [11].

Cell lines used in the present study are also used in other previous reports. Among them, the greatest effect belongs to hyperatomarin with an IC_50_ of 0.86 μM against MDA-MB-231 cells [69]. Hyperforin has an IC_50_ of 12 μM [70] for these cells and of 21 μM for K-562, while its synthetic derivatives have a lower value and the dicyclohexylammonium salt of the compound—only 3.2 μM [71,72]. The spirocyclic hyperbeanols B and D from *H. beanii* N. Robson had IC_50_ values of 17 and 21 μM against K-562 cells, respectively [73]. Maculatoquiones A–D and erectquione A, simple phloroglucinol derivatives from *H. maculatum* Crantz, had IC_50_ values in the range of 21–77 μM [40]. 7-*Epi*-clusianone, a BPAP, and elegaphenone, a benzoylphloroglucinol, from *H. elegans* Stephan ex Willd. had IC_50_ values ranging from 9.8 to 13.6 μM and they were lower when the solvent was EtOH (unpublished data) in comparison to DMSO [41]. Other phloroglucinols and xanthones with a phloroglucinol core [74] from *H. henryi* subsp. *uraloides* (Rehder) N. Robson [75], *H. ascyron* [76] and *H. chinense* L. [77] had no effect on K-562 cells.

Benzophenones and a xanthone with a phloroglucinol core were tested against HL-60, HL-60/DOX and K-562. Gentisein was relatively most active with an IC_50_ of 88.3 to 102 μM; the other compounds had a very weak cytotoxicity with an IC_50_ of 120 to >400 μM. Their mechanism of action is the induction of apoptosis (found in HL-60 cells). All substances increase the sensitivity of HL-60/DOX to anthracyclines, i.e., they overcome the resistance [78].

The PPAP hyperisampsin J from *H. sampsonii* induces apoptosis in HL-60 cells and has an IC_50_ of only 0.56 μM, lower than that of cisplatin, and an SI value of 2.67. Hyperisampsins M and K have IC_50_ values of 1.4 μM and 1.7 μM [53], the latter value be-longs also to hypercalin C, a more simple phloroglucinol derivative, while an adaman-tane PPAP (both from *H. henryi* H. Lév. ex Vaniot) has an IC_50_ value of 24 μM [79]. A much weaker effect is attributed to the homo-adamantane PPAPs from *H. cohaerens* N Robson with IC_50_ values > 40 μM [80]. Phloroglucinols from *H. beanii*, *H. hookerianum*, Wight ex Arn. and *H. uralum* Buch.-Ham. ex D. Don are not active towards that cell line [73,81,82]. Intermediate activity is attributed to benzoylphloroglu-cinols, BPAPs, PAPs, etc., from *H. sampsonii* [11,54,83,84], *H. cohaerens* [84,85,86], *H. attenuatum* [3,87], *H. ascyron* [88,89], *H. uralum* [82], *H. perforatum* [14,52], *H. henryi* [90] etc. [75]. In some cases, the IC_50_ value, e.g., 3.3 μM of hyphenrone R from *H. henryi*, is close to that of the positive control, e.g., cisplatin [91], and in others there is a high selectivity (SI 7, relative to the normal cells Beas-2B) for a phloroglucinol derivative from *H. attenuatum* [92]. Some structural differences that probably cause great differences in activity are described by Zhu et al. [93].

Phloroglucinol derivatives from other plant taxons (*Callistemon*, *Myrtaceaea*, etc.) and even fungi have been tested on these cell lines and range in their effects from strong (IC_50_ of 3 μM) to no cytotoxicity, and some of them induce apoptosis [94,95,96]. 

MIC values of single compounds, e.g., from Chinese St. John’s wort, range of 0.8–16 μM, but these include antimicrobial compounds other than phloroglucinols, such as naphtodianthrones (hypericins), xanthones, flavonoids and benzopyrans [97].

Individual phloroglucinol derivatives from *Hypericum* have varying antimicrobial activity—from exceptionally high to weak or lacking. To the best of our knowledge, hyperforin, as the main antibacterial principle from St. John’s wort, has demonstrated the lowest MIC value of 0.1 mg/L (0.19 μM) against *S. aureus*, *Sarcina lutea* [5] and *Corynebacterium diphtherie* [6]. Previous research has reported similar low MIC values (0.2 μM), equivalent to or lower than those exhibited by the control antibiotics, for dimeric acylphloroglucinols (drummondins) from *H. drummondii* for *Bacillus subtilis* [98]. Additionally, as mentioned, olympicin A (compound **2** in the present study) has MIC values ranging from 0.5 to 1 mg/L against MDR and epidemic *S. aureus* and MRSA. It was more active than the control antibiotics [39] and also inhibited the efflux from NorA multidrug efflux pumps in *S. aureus* [55].

However, other BPAPs or simpler phloroglucinol derivatives have shown higher MIC values. Hyperforin has an MIC of 1 mg/L against MRSA, *E. faecalis* and other Gram-positive bacteria. However, against *S. aureus*, the same compound has shown an MIC of 50 mg/L and against Gram-negative bacteria, such as *E. coli* and fungi, an MIC of 400 mg/L [5]. Other phloroglucinols from *H. annulatum* [99], *H. drummondii* [98], *H. pseudopetiolatum* [2], *H. andinum* Gleason, *H. laricifolium* Juss., *H. brevistylum* Choisy, *H. silenoides* Juss., *H. brasiliense* Choisy and *H. sampsonii* [98] have MICs of 0.78–8 μM (and similar values expressed as mg/L) against *S. aureus*, including a norfloxacin resistant MDR strain [24]. Against other bacterial strains, such as MRSA, *Staphylococcus epidermidis*, *Micrococcus luteus*, *Bacillus cereus*, *Streptococcus mutans* and *Mycobacterium smegmatis*, phloroglucinols from these species exhibited MIC values of 3–14 μM and 1.25–128 mg/L [2,4,39]. The MIC values against fungi, such as *Candida albicans*, *Aspergillus niger*, *Trichophyton mentagrophytes* and *Cryptococcus neoformans*, were as low as 2 to 33 mg/L [2]. 

Recent research has shown that dimeric acylphloroglucinols from *Hypericum japonicum* Thunb. also have very low MIC values of 0.8 to 86 μM, and the lowest MICs were against *E. coli*, *E. faecalis* and *Salmonella typhymurium* while the lowest MIC against *S. aureus* was 1.7 μM [18]. The lowest MIC_50_ value (50% of bacterial growth measured by absorption) of seco-spirocyclic PPAP derivatives from *Hypericum longistylum* Oliv. against *S. aureus* subsp. *aureus* was 11.2 μM [17]. Some BPAPs from *H. scabrum* had a weak effect on MRSA and *B. subtilis* with MIC values of 320 μM but no effect on MDR *Pseudomonas aeruginosa* [68]. The lowest MIC of dimethylated acylphloroglucinol meroterpenoids from *H. elodeoides* Choisy against oral bacteria, including *Streptococcus* spp., was 6.25 μg/mL [20]. Prenylated acylphloroglucinols with menthane moieties from *H. ascyron* had ranges of MIC values of 1–8 μM for *S. aureus* and MRSA, 1–4 μM for *B. subtilis* and > 64 μM for *E. coli* [19]. 

Examples of MIC values of phloroglucinols from plant species other than *Hypericum* are 0.25–8 mg/L towards different strains of *S. aureus*, *S. pyogenes*, etc., and 8 mg/L against *E. coli* [4]. When we consider our results and those in the literature, it can be concluded that, similarly to *Hypericum* extracts, single phloroglucinols are inactive or have little activity against Gram-negative bacteria because of their outer membrane, but there are many exceptions.

Therefore, in comparison to the results from the literature, the phloroglucinol derivatives in the present study showed antibacterial effects in the low micromolar ranges and a very significant activity against *S. aureus* with MIC values in the range of 0.78–2 mg/L. Compound **5** also had an excellent effect against MRSA with an MIC value of 1 mg/L.

Research on the effect of *Hypericum* phloroglucinols and bacterial biofilm is not abundant. Five phloroglucinol derivatives from four species had MBIC values against MRSA of 3.9–7.8 mg/L, but the MBIC value was lower than the respective MIC for only one of them. Their MBICs against the biofilms from *S. aureus* and *S. epidermidis* were 2–7.8 mg/L, and most of these values were lower than the respective MIC values [31]. A dicyclohexylamonium salt of hyperforin and its analogue had MBICs on the biofilm of a MRSA clinical isolate of 25–37.5 mg/L and on the biofilm of *S. aureus* and *E. faecalis*—25 to 150 mg/L, but they were much higher than their MICs of 1–4 mg/L [32]. Therefore, the data show that compound **5** has a very significant antibiofilm activity with an MBIC value of 0.5 mg/mL (½ MIC). Compound **3** also showed a strong antibiofilm effect with an MBIC value of 4 mg/mL (again ½ MIC) (Table 9).

AMR is a natural phenomenon, which is accelerated by the selective pressure exerted by the widespread use and abuse of antibiotics in humans and animals. An aggravating factor is the horizontal transfer of resistance genes between bacteria, as well as from non-pathogenic bacteria from the environment to clinically significant strains. 

As we know, MRSA, which is resistant to β-lactam antibiotics or is even MDR, is a dominant type of *S. aureus* infection, causes from mild skin infections to fatal septicemia and is one of the most lethal human pathogens [30,100]. These high rates imply that standard prophylaxis with first-line drugs for surgery will have limited effect, and treatment for severe infections must rely on second-line drugs in many countries. Second-line drugs for *S. aureus* are more expensive and have severe side effects, the monitoring for which increases costs even further [101]. 

The AMR (including for immune factors of the host) of MRSA and biofilm forming bacteria, such as other *Staphylococcus* spp., *E. coli* and *P. aeruginosa* [31], increases to several hundred times in biofilms, mainly because of decreased drug permeability and appearance of dormant persisters [102]. The reason for the selection of MRSA for biofilm research in this study is, as we have already emphasized, MRSA biofilms are persistent, colonize catheters and other devices and are significantly lethal to patients with (necrotic) wounds [29,30,100,103].

As mentioned, cancer is the second leading cause of death globally accounting for an estimated 9.6 million deaths, or one in six deaths, in 2018. Lung, prostate, colorectal, stomach and liver cancer are the most common types of cancer in men, while breast, colorectal, lung, cervical and thyroid cancer are the most common among women [28]. 

Natural products are markedly effective in infectious and oncological diseases because their targets are major crossroads in signaling pathways in pathogens or cancer cells [104]. Sixty percent of the drugs currently used to treat and prevent cancer are derived from natural products from terrestrial and marine sources or microorganisms since the de novo chemical synthesis of chiral molecules is not chemically justified [105,106]. As the usual time from the discovery of a lead structure to its approval for use is 8–15 years, the next few years will be critical for the discovery of new drugs of natural origin with neoplastic effects or with the ability to bypass AMR [107,108].

Regarding compounds **1** and **3**, this is the first case of isolation of these known compounds from *H. olympicum*. Olympicins B–E, which have been found to accompany olym-picin A (**2**) [39], were not isolated in this study, pointing out that they could be oxidized artifacts of isolation procedures. Compound **3** (only one tautomer or both) has been detected or isolated so far from *H. tetrapterum* Fr., *H. undulatum* Schousb. ex Willd., *H. kouytchense* H. Lév., *H. polyphyllum* Boiss ex Bal. [67], *H. perforatum* [109] and *H. maculatum* [40].

Compounds **1**–**3** have already been studied for cytotoxic activity. While polar extracts from *H. olympicum* do not possess cytotoxic activity [110], moderate cytotoxicity was observed for **2** with IC_50_ values of 8.9–12.5 μM against human tumor cell lines [55]. Also, compounds **1** and **2** showed inhibitory activity at 25 mg/L against the growth of a panel of tumor cell lines in the range of 45–84% for **2** and 48–82% for **1** [42]. Compound **3** exerted IC_50_ values of 5.3, 5.7 and 21.3 μM on human leukemia cell lines SKW-3, BV-173 and K-562, respectively. The panel of cell lines in the current study also included K-562 and the IC_50_ value turned out to be significantly lower (3.17 μM). This time, the compound had greater cytotoxic activity generally and in particular cell lines. The explanation may be given by the fact that in the previous work [40] the compound was dissolved in DMSO. The substance has unstable bi-dicarbonyl core-like hyperforin and it is most likely that it was partially degraded by the solvent. The compound is extremely sensitive to air and light too [109].

Compound **1** has also been examined for antibacterial activity and it inhibited the growth of MRSA with a low MIC_50_ value (1.14 mg/L). It was even more potent than **2**, which had an IC_50_ value of 1.80 mg/L [42]. Both compounds were again not active against Gram-negative bacteria [31,39]. The potential to inhibit bacterial biofilm production of both compounds was assessed. The MBIC values ranged from 1.95 mg/L to 3.91 for **1** (which was at or below its respective MIC and minimal bactericidal concentration values) and from 3.91 to 7.81 μg/mL for **2** [31].

In regard to the structure–activity relationship, the increased lipophilicity of a metabolite increases cytotoxic activity as the ability for penetration through the plasma membrane of the eukaryote cell (e.g., a tumor or protozoan cell) is improved [111,112]. Hence, prenylation is beneficial for antineoplastic activity. However, concerning antibacterial activity, it has also been suggested that additional prenyl groups would probably limit aqueous solubility, and thus decrease intracellular concentrations and antibacterial activity [113] since the synthetic increase of a geranyl ether group to farnesyl is more detrimental than a decrease [114]. Indeed, for this study, we can observe that generally PPAPs with three prenyl groups and higher molecular weight and log Po/w have stronger cytotoxic properties than geranylated simple acylphloroglucinols. Nevertheless, the latter, which are more soluble, have a higher antibacterial activity and higher gastrointestinal absorption. The mechanism of cytotoxicity of **2** and analogues is not completely known, but authors [42,55] suggest that since all of the most active compounds have a hydrophobic part (a hydrocarbon and an acyl functionality), capable of membrane interaction, and a hydrophilic diphenolic moiety, it is probable that their target is located in the cell wall. Our research showed that if we compare the BPAPs of hyperforin-type **3** and **5** with the structurally similar **2** and **1**, they also have a hydrophobic part and a core, which is hydrophilic, but with a slightly bigger polar surface area and less lipophilic. BPAPs as whole molecules differ in that they have a bigger molecular weight and TPSA, greater lipophilicity and less solubility. Still, they (especially **5**) showed the same significant activity. 

Further research may include antimicrobial testing of the methylates of **4**. It is interesting whether they would exhibit a decrease of activity compared to **4** since a study has reported that PPAPs with covalent block of the tautomeric equilibrium showed more than a 10-fold decrease of 5HT reuptake inhibition compared to hyperforin, suggesting that this modification is detrimental to its activity [47]. Initial data are collected and studies to establish the mechanism of antibacterial activity of the tested compounds are forthcoming.

## 4. Materials and Methods

### 4.1. General Experimental Procedures

Optical rotations were recorded at 589 nm on a Rudolph Research Analytical Autopol VI (Hackettstown, NJ, USA) polarimeter equipped with a TempTrol™ 100 mm sample cell. UV spectra were measured in LC/MS grade MeOH on a Biochrom Libra S70 (Cambridge, UK) spectrophotometer. All ^1^H, ^13^C and 2D NMR spectra were recorded at 300 K on a Bruker AVANCE II+ 600 (Germany) spectrometer (operating at 600.13 MHz for ^1^H and 150.90 MHz for ^13^C) with BBO probe. Spectra were taken in CDCl_3_ or CD_3_OD, referenced against TMS. Shift values (*δ*_H_ and *δ*_C_) are always given in ppm and *J* values in Hz. The standard pulse sequence and phase cycling were used for MQF-COSY, HSQC, HMBC and NOESY experiments. The NMR data were acquired and processed using Bruker Topspin 3.0 software. HRESIMS and MS/MS spectra were acquitted on a ThermoFisher Scientific Q Exactive Plus (Bremen, Germany) mass spectrometer, supplied with a HESI-II source. Column chromatography (CC) was carried out over sorbents Diaion HP-20 (65 × 100 mm) or MCI gel (Supelco). Low pressure liquid chromatography (LPLC) was done over a glass column (25 × 370 mm) filled with Merck LiChroprep^®^ (Darmstadt, Germany) C18 (0.04–0.063 mm 40–63 μm, 4 × 16 cm) at 10 mL min^−1^. Semi-preparative (SP) HPLC was performed on a Waters (Milford, MA, USA) Breeze2 high-pressure binary gradient system consisting of a binary pump model 1525EF, manual injector 7725i and an UV detector model 2489 or on Young Lin 9100 (Hogye-dong, Anyang, Korea), consisting of vacuum degasser YL 9101, quaternary pump YL 9110, manual injector 7725 and YL 9160 PDA detector. The column used for SP-HPLC were either Kromasil C18 (250 × 10 mm, 5 μm, flow 5 mL.min^−1^) or Kromasil C18 (250 × 21.2 mm, 5 μm, flow 18 mL·min^−1^) purchased from Eka Chemicals AB (Bohus, Sweden).

### 4.2. Plant Material

The flowering aerial parts of *Hypericum olympicum* L. were collected from wild habitats near Momchilgrad and Krumovgrad (East Rhodopi Mountain) in June 2015. They were authenticated by Assoc. Prof. Paraskev Nedialkov. The voucher specimen (No. 1346) was deposited at the herbarium of the Institute of Biodiversity and Ecosystem Research (IBER) at Bulgarian Academy of Sciences (BAS).

### 4.3. Extraction and Isolation

The fresh plant material was dried in shade at room temperature for 10 days and powdered to give 279.5 g dry mass. The powdered plant material was extracted by percolation with CH_2_Cl_2_-MeOH (4:1) (12 L) at room temperature. The solvent was evaporated under vacuum in rotary evaporator to give 46.9 g a dark green wax-like residue. It was suspended in 180 mL MeOH, sonicated for 15 min and then 20 mL of H_2_O was added. The mixture was passed through pre-soaked with MeOH-H_2_O (9:1) Diaion HP-20 (50 g). The sorbent was subsequently washed with 3 L MeOH-H_2_O (9:1) mixture, 1 L MeOH and with CH_2_Cl_2_ until decolorization of the solvent. The resulted solutions were separately evaporated under vacuum. The aq. methanol residue (42.8 g) divided into four equal portions. Each portion (*ca*. 10 g) was separately subjected on a N_2_ driven (*ca*. 1 atm) CC on MCI gel column and was eluted with mixtures of MeOH-H_2_O (50:50→100:0, step of 10, each 1.5 L, fraction vol. 500 mL). The individual fractions were combined based on LC-MS analysis into five pooled fractions (I-V). An aliquot (*ca*. 1 g) of fraction IV was subjected on LPLC separation and eluted with MeOH-H_2_O (80:20 → 100:0 with 0.02% H_3_PO_4_, step of 5, each 1 L, fraction vol. 50 mL). The procedure was repeated in triplicate. The fractions were combined into twelve pooled fractions (A-L) according to their composition. Fraction E (193 mg) was subjected to isocratic SP-HPLC and eluted with MeOH-H_2_O (75:15 with 0.02% H_3_PO_4_) to give **1** (9.7 mg) and **2** (41.7 mg). Fraction H (1.17 g) was separated by isocratic SP-HPLC with MeOH-H_2_O (86:14 with 0.02% H_3_PO_4_) to give **3** (108.5 mg) and **4** (62.4 mg). SP-HPLC of F (105 mg) were chromatographed by SP-HPLC with eluent MeOH-H_2_O (80:20 with 0.02% H_3_PO_4_) to give **5** (28.6 mg).

The removing of H_3_PO_4_ was done by the following procedure. Briefly, the solutions containing H_3_PO_4_ were concentrated in vacuo until at least 90% of the CH_3_CN or MeOH were removed. Subsequently, the concentrated solutions were filtered over Diaion HP20SS and washed with H_2_O. Finally, the solutes were recovered out of the resin by washing with MeOH.

### 4.4. Methylation of Compound **4**

Compound **4** (40 mg) was treated with CH_2_N_2_ according to the procedure given in the literature [40]. The mixture of the methylates of both keto-enol tautomers was subjected to SP-HPLC separation with CH_3_CN-H_2_O (85.5:14.5) as eluent and gave **4a** (9.2 mg) and **4b** (19.1 mg).

### 4.5. New Compound Data

#### 4.5.1. (E)-1-[2-{(3,7-dimethylocta-2,6-dien-1-yl)oxy}-4,6-dihydroxyphenyl]-2-methylpropan-1-one (1)

Colorless oil; UV (MeOH) *λ*_max_ (log ε) 230 (3.82), 290 (4.07) nm; (+NaOAc) 323 nm; (+AlCl_3_/HCl) 311, 362 nm; ^1^H and ^13^C NMR: See Table 1; HRESIMS *m/z* 331.1913 [M−H]^−^ (Calculated for C_20_H_27_O_4_, 331.1904); MS/MS (NCE30) *m/z* 152.011 (24%), 194.058 (100%), 287.202 (12%).

#### 4.5.2. Olympiforin A or Mixtures of Keto-Enol Tautomers (1R,5R,7S,8R)-4-hydroxy-5,8-dimethyl-3,7-bis(3-methylbut-2-en-1-yl)-1-(2-methylbutanoyl)-8-(4-methylpent-3-en-1-yl)bicyclo[3.3.1]non-3-ene-2,9-dione and (1R,5S,6R,7S)-4-hydroxy-1,6-dimethyl-3,7-bis(3-methylbut-2-en-1-yl)-5-(2-methylbutanoyl)-6-(4-methylpent-3-en-1-yl)bicyclo[3.3.1]non-3-ene-2,9-dione (**4**)

Colorless oil; ${\left[\alpha \right]}_{D}^{20}$ +20.97 (c 0.102, MeOH); UV (MeOH) *λ*_max_ (log ε) 295 (3.84) nm; ^1^H and ^13^C NMR: See Table 2 and Table 3; HRESIMS *m/z* 497.3622 [M+H]^+^ (Calculated for C_32_H_49_O_4_, 497.3625); MS/MS (NCE30) *m/z* 441.300 (8%), 373.237 (1%), 293.175 (44%), 223.096 (74%), 163.148 (8%), 69.071 (29%).

#### 4.5.3. (1R,5R,7S,8R)-4-methoxy-5,8-dimethyl-3,7-bis(3-methylbut-2-en-1-yl)-1-(2-methylbutanoyl)-8-(4-methylpent-3-en-1-yl)bicyclo[3.3.1]non-3-ene-2,9-dione (**4a**)

Colorless oil; ^1^H and ^13^C NMR: See Table 2 and Table 3; HRESIMS *m/z* 511.3781 [M+H]^+^ (Calculated for C_33_H_51_O_4_, 511.3782), 533.3598 [M+Na]^+^ (Calculated for C_33_H_50_O_4_Na, 533.3601); MS/MS (step NCE 10, 30, 60) *m*/*z* 455.315 (2%), 307.190 (17%), 251.128 (20%), 233.117 (7%), 163.148 (5%), 69.071 (47%).

#### 4.5.4. (1R,5S,6R,7S)-4-methoxy-1,6-dimethyl-3,7-bis(3-methylbut-2-en-1-yl)-5-(2-methylbutanoyl)-6-(4-methylpent-3-en-1-yl)bicyclo[3.3.1]non-3-ene-2,9-dione (**4b**)

Colorless oil; ^1^H and ^13^C NMR: See Table 2 and Table 3; HRESIMS *m/z* 511.3780 [M+H]^+^ (Calculated for C_33_H_51_O_4_, 511.3782), 533.3601 [M+Na]^+^ (Calculated for C_33_H_50_O_4_Na, 533.3601); MS/MS (step NCE 10, 30, 60) *m*/*z* 307.190 (37%), 251.128 (88%), 233.118 (23%), 163.148 (11%), 69.071 (74%).

#### 4.5.5. Olympiforin B or (1R,5S,6R,7S)-4-hydroxy-5-isobutyryl-6-methyl-3,7-bis(3-methylbut-2-en-1-yl)-6-(4-methylpent-3-en-1-yl)bicyclo[3.3.1]non-3-ene-2,9-dione (**5**)

Colorless oil; ${\left[\alpha \right]}_{D}^{20}$ + 9.8 (c 0.116, MeOH); UV (MeOH) *λ*_max_ (log ε) 289 (3.91) nm; ^1^H and ^13^C NMR, Table 4; HRESIMS *m*/*z* 469.3313 [M+H]^+^ (Calculated for C_30_H_45_O_4_, 469.3312); MS/MS (NCE 10) *m/z* 345.206 (23%), 413.268 (69%), 277.143 (100%).

### 4.6. Cell Lines and Culture Conditions

The group of human cell lines used in this study were represented by the tumor cell lines MDA-MB (breast carcinoma), EJ (urinary bladder carcinoma), K-562 (chronic myeloid leukemia), HL-60 (acute pre-myeloid leukemia), HL-60/DOX (multi-drug resistant HL-60 cell line, with over expression of MRP-1) and the non-tumorigenic cell lines HEK-293 (embryonic kidney cells) and EA.hy926 (vascular endothelial cells). EJ, HL60 and EA.hy926 were purchased from American Type Culture Collection (ATCC) (Manassas, Virginia). The other cell lines were purchased from the German Collection of Microorganisms and Cell Cultures (DSMZ GmbH, Braunschweig, Germany). HL-60/DOX cell lines were generated by periodically treating HL-60 cells with anthracyclins. The conditions in which the cell culture flasks were maintained were as follows: 37 °C in an incubator “BB 16-Function Line” Heraeus (Kendro, Hanau, Germany) with 5% CO2 humidified atmosphere. The growth medium was 90% RPMI-1640, supplemented with 10% FBS and 2 mM L-glutamine. MDA-MB-231, EJ, HEK-293 and EA.hy926 cells were maintained as adherent monolayer cultures while all the other (leukemic) cell lines were grown as suspension type cultures. Cells were maintained in log phase: the adherent cell lines were subjected to trypsinization 1–2 times a week and the leukemic cell lines were passaged 2–3 times a week by removing of part of the cellular suspension and substituting it with fresh medium.

### 4.7. In Vitro Cytotoxicity Determination

The evaluation of cytotoxic activity was conducted by MTT test as described previously [115]. The median inhibitory concentration (IC_50_) values which represent the half-maximal concentration values were calculated from sigmoidal concentration–response curves, using non-linear regression analysis (Curve fit, GraphPad Prizm 5.0 software, GraphPad Software Inc., San Diego, CA, USA). The selectivity index was calculated as the ratio of the IC_50_ of a non-cancerous line (HEK-293 or EA.hy926) and the average IC_50_ of all chemosensitive tested tumor cell lines (*x*). Therefore,$\;SI=\frac{IC_{50}^{\mathrm{HEK}\text{-}293}}{IC_{50}^{x}}$. The resistance index (RI) was calculated as the ratio of IC_50_ of the multi-drug resistant HL-60/DOX cell line and the IC_50_ of its parent cell line: $RI=\frac{IC_{50}^{\mathrm{HL}\text{-}60/\mathrm{DOX}}}{IC_{50}^{HL\text{-}60}}$.

### 4.8. Antiproliferative Test

The test was performed according the methodology of National Cancer Institute (NCI), USA [116]. Briefly, cells were seeded into 96-well plates in the same way as for an MTT test and after 24 h, the absorption of one plate of each cell line was measured as in the MTT assay described above, to represent a measurement of the cell population at the time of drug-addition (T_z_). The rest of the plates were treated with the tested compounds and the experiment for them continued as the MTT-method. Using the absorbance measurements [(T_z_, untreated control growth (C) and test growth in the present of the compounds (T_i_)], the percentage growth is calculated at each of the tested concentrations of the compounds. Percentage growth inhibition (*GI*) is calculated as $GI\%=\frac{T_{i}-T_{z}}{C-T_{z}}\times 100$ for concentrations for which *T*_*i*_ ≥ *T*_*z*_. For concentrations in which T_i_ < T_z_ it is calculated as $GI\%=\frac{T_{i}-T_{z}}{T_{z}}\times 100$. Two dose parameters are calculated for each experimental agent. Growth inhibition of 50% (GI_50_) is calculated from $\frac{T_{i}-T_{z}}{C-T_{z}}\times 100=50$. This is the drug concentration resulting in a 50% reduction in the net protein increase (as measured by MTT-test) in control cells during the drug incubation. The drug concentration resulting in total growth inhibition (TGI) is calculated from T_i_ = T_z_.

### 4.9. Western Blot

The protocol for protein immunoblots that we used began with MDA-MB-231 and HL60/DOX (2 × 10^6^ cells) treated with all the compounds isolated in this study, dissolved in ethanol (equieffective concentrations multiple of IC_50_ − IC_50_ and ½ IC_50_, respectively) and incubated for 24 h at the aforementioned conditions. Next, the samples were lysed on ice with a buffer containing 100 mM Tris–HCl at pH 8.0, 4% SDS, 20% glycerol, 1% Na_3_VO_4_, 200 mM dithiothreitol (DTT) (Sigma-Aldrich, St. Louis, MO, USA) and protease inhibitor (Roche, Basel, Switzerland). Then the cell lysate was boiled for 15 min and centrifuged at 13,000 rpm for 10 min at 4 °C. Before DTT addition, aliquot samples of 10 μL were taken from the lysates. Their protein concentrations were determined using a BCA protein assay kit (ThermoFisher Scientific, Waltham, MA, USA) after dilution 5× with H_2_O. Subsequently, the samples were subjected to denaturing SDS-PAGE electrophoresis and then transferred to PVDF membrane (Santa Cruz Biotechnology, USA). After this, the blots were fixed with blocking solution (5% *w*/*v* nonfat dried milk powder, dissolved in TBS) and incubated with specific primary antibodies: mouse procaspase-9 (sc-17784), mouse Bcl-2 (C-2, sc-7382) and mouse β-actin (sc-8432). The incubation was either overnight at 4 °C or for 150 min at room temperature. After washing, the membranes were incubated for 45 min with horseradish peroxidase conjugated secondary goat anti-mouse antibody (sc-2005) or mouse IgG kappa antibody (sc-516102) (Santa Cruz Biotechnology, Santa Cruz, CA, USA). After washing again, the blots were developed with WesternSure^®^ Premium Chemiluminiscent substrate (LI-COR, Inc., Lincoln, NE, USA) and imaged with C-DiGit Blot Scanner (LI-COR, USA). Control on the even application of the samples was conducted by comparison of the levels of the ubiquitous protein *β*-actin in the separate samples. The relative density of the bands of the treated samples (compared to *β*-actin) was normalized as percentage of the untreated control. This densitometry analysis of the blots was calculated by Quantity One software version 4.6.9 (Bio-Rad Laboratories, Hercules, CA, USA). Statistical analysis utilized Student’s *t*-test with *p* ≤ 0.05 set as significance level.

### 4.10. Bacterial Strains and Growth Conditions

The Gram-positive bacterial strains that were examined for susceptibility were: *Staphylococcus aureus* (ATCC 29213, American Type Cell Culture Collection, Manassas, VA, USA), *Staphylococcus aureus*—MRSA (NBIMCC 8327-resistant to methicillin and oxacillin, National Bulgarian Collection for Industrial Microorganisms and Cell Cultures, Sofia, Bulgaria), *Enterococcus faecalis* (ATCC 29212) and *Streptococcus pyogenes* (SAIMC 10535, Collection of the Stephan Angeloff Institute of Microbiology, Sofia, Bulgaria). The only Gram-negative bacterial strain used was *Escherichia coli* (ATCC 35218). Bacteria were maintained in Trypticase Soy Broth (TSB, Himedia, India) at 37 °C, aerobic conditions. For the experiments brain heart infusion (BHI) broth (#M210 HiMedia, Mumbai, India) was used for *E. faecalis* and *S. pyogenes* and Mueller Hinton broth (MHB, #M0405B) from Thermo Scientific-Oxoid (Basingstoke, UK)—for all other strains.

### 4.11. MIC Determination by Broth Microdilution Method (BMD)

BMD was performed according to the following protocol of ISO 20776/1-2006 [117]. A bacterial suspension with density of 10^8^ CFU/mL (0.5 McFarland standard, OD600) was prepared from an overnight grown liquid bacterial culture homogenized over vortex and brought to a working bacterial suspension (WBS) with concentration of 5×10^5^ CFU/mL by being diluted 200× with MHB or BHI broth and homogenized over vortex again. Two-fold serial dilutions of compounds **3**–**5** was prepared in 96-well round bottom plates to a volume of 50 μL and final range of concentrations 100–0.2 μg/mL. Stock solutions were dissolved in ethanol and MHB was used as diluent and as test for absence of contamination. Three wells were left as blank samples and 6 wells were left for control samples. An equivalent volume (50 μL) of the WBS was inoculated to each well in the aforementioned plates achieve final bacterial density of 5 × 10^4^ CFU/mL with the exception of the blank wells. After 18–24 h incubation under aerobic standardized conditions at 37 °C the plates were examined. The growth in the positive control wells was checked for sufficient growth and the MIC was determined manually as the lowest concentration that completely prevented or inhibited visible bacterial growth, as detected by the unaided eye, expressed in μg/mL. Antibiotics dissolved in sterile distilled water were used as referent positive controls. They were penicillin G (PEN, #B0500000, Merck KGaA, Darmstadt, Germany) for *E. faecalis* and *S. pyogenes*, gentamicin (GEN, #15750-037, 50 mg/mL, Gibco, Paisley, UK) for all bacteria except *S. pyogenes* and ciprofloxacin (CIP, Ciproflav:10 mg/mL, Polfa S.A. Warsaw Pharmaceutical Works, Starogard Gdanski, Poland) for *E. faecalis*. The range of concentrations were 0.004–2 mg/L for PEN, 0.008–4 mg/L for GEN and 0.005–2 mg/L for CIP. The requirements of EUCAST (European Committee on Antimicrobial Susceptibility Testing) for their MICs were followed for discussing the results [118]. In the end the plates which contained active agents were subjected to assessment of the dehydrogenase activity of the bacteria.

### 4.12. Assessment of the Cell Redox (Dehydrogenase) Activity

Dehydrogenase activity was measured according to the protocol of [119] with minor modifications using the MTT dye again since it is reduced by the membrane located bacterial enzyme NADH: ubiquinone reductase to formazan crystals. A total of 10 μL MTT (5 mg/mL in PBS) was added to each well at the end of the incubation period after reading the BMD assay and mixed thoroughly. The plate was incubated at the same conditions for 1 h (4–5 h for *S. pyogenes* as this strain had weak growth and dehydrogenase activity). Then the medium was removed and the formazan crystals were dissolved by addition of an equivalent volume (100 μL/well) of 5% formic acid solution in 2-propanol. The absorbance was measured at 550 nm (Absorbance Microplate Reader Lx800, Bio-Tek Instruments Inc., Winooski, VT, USA) with the lid against a blank solution containing respective volumes of MHB, MTT and solvent. Dehydrogenase activity was calculated as percentage of the activity of the normalized control.

### 4.13. Biofilm Assay of MRSA

Equieffective concentrations of compounds **3** and **5** multiples of MICs (MIC, ½ MIC, ¼ MIC, 1/8 MIC, 1/16 MIC and 1/32 MIC) were prepared in 96-well polystyrene flat bottom tissue culture plates in BHI containing 2% glucose (*w*/*v*) to a final volume of 100 μL/well. The bacterial inoculum was prepared as described above for the BMD. The next steps were addition of an equivalent volume (100 μL) of MRSA solution to each well and incubation for 18–24 h in the same manner as in the MIC protocol. After that an optimized protocol was applied for the visualization of the biofilm [120]. Firstly, the supernatant was removed, and planktonic cells were removed by washing the wells carefully 3× with 200 μL/well PBS. Then, fixing of the remaining cells was performed by incubation with 200 μL/well methanol for 15 sec at room temperature. The methanol was removed and the plate was dried by air for 5–10 s. Next, 50 μL of 2% Hucker crystal violet was poured into the wells and excess stain was rinsed off by running tap water. After that, the cells were air dried and the biofilm formation in the wells was documented microscopically (40×, Inverted trinocular biological microscope model BIB-100 and digital camera B-CAM16, application software B-View, Boeco, Hamburg, Germany). Then the dye bound to biofilm in the wells was re-solubilized with 160 μL 33% acetic acid and the absorbance of each well was measured at 550 nm with lid (Appendix A). The minimum biofilm inhibition concentration (MBIC) is the lowest concentration of an antimicrobial agent that results in no detectable biofilm growth [121] while the half-maximal MBIC (MBIC_50_) is the lowest concentration of the tested compounds that leads to 50% inhibition on the biofilm formation [31], as calculated by sigmoidal concentration–response curves, using non-linear regression analysis (Curve fit, GraphPad Prizm 5.0 software, GraphPad Software Inc., San Diego, CA, USA). MBIC were assessed visually and MBIC_50_ were calculated by GraphPad Prism software with a mathematical model for a dose–response relationship (variable slope) after normalization of the data and logarithmic transformation of the applied concentrations (X-data).

### 4.14. Physiochemical Properties and General Computational Methodology, ADME and Drug-Likeness Estimation

In silico ADME and drug-likeness was evaluated using the web tool SwissADME as we have already described elsewhere [57]. 

### 4.15. Statistics

At least 4 wells were treated with each concentration of the compounds for the MTT assay. Two to three wells per concentration were used for the BMD, the cell redox and the biofilm assay, and at least four wells were left for control and at least three for the blank samples for each assay. One-way ANOVA was utilized for the experiment measuring the redox (metabolic) activity (Appendix A).

## 5. Conclusions

Five phytochemicals were isolated from the taxon *Hypericum olympicum*—two new bicyclo[3.3.1]nonane type polyprenylated acylphloroglucinols derivatives as well as three known prenylated phloroglucinols and their structures were elucidated. They were shown to possess significant cytotoxic activity in the micromolar range. Most of the com-pounds had good selectivity in terms of normal versus tumor cell lines. The mitochondrial pathway of apoptosis is the most likely mechanism of cytotoxic activity. The bilycyclic polyprenylated acylphloroglucinol derivatives (BPAPs) showed a moderate effect against *E. faecalis* and *S. pyogenes* but a very significant activity against *S. aureus* with MIC values in the range of 0.78–2 mg/L. Olympiforin B (**5**) also had an excellent effect against MRSA with an MIC value of 1 mg/L and a very significant antibiofilm activity on that strain with an MBIC value of 0.5 mg/L. The drug-likeness evaluation showed that the BPAPs had stronger cytotoxic properties than the geranylated simple acylphloroglucinols, but the latter had higher antibacterial activity and higher gastrointestinal absorption. The results from this study are another confirmation that PPAPs from *Hypericum* are a promising new class of leading molecules for obtaining drugs and food additives with pleiotropic activity.

## Figures and Tables

**Figure 1 plants-12-01500-f001:**
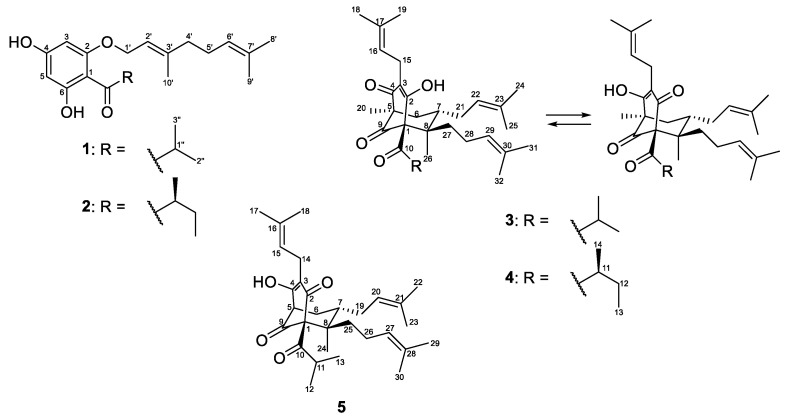
The structures of the isolated compounds **1**–**5**.

**Figure 2 plants-12-01500-f002:**
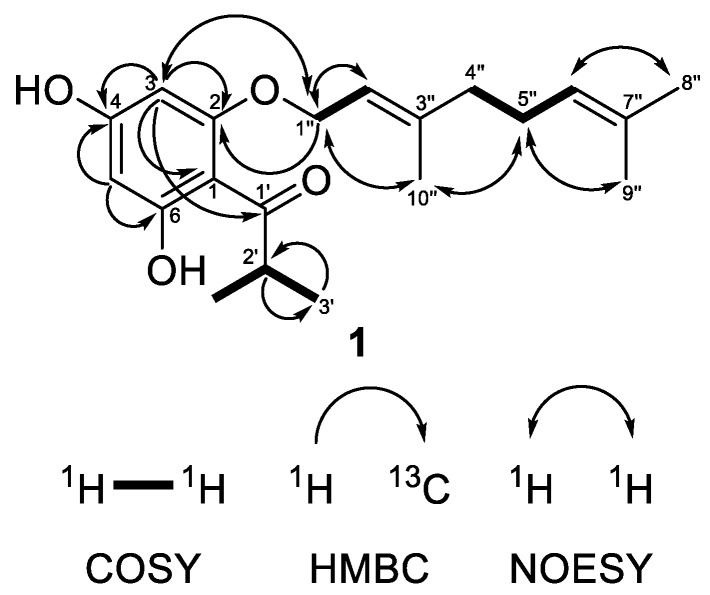
Selected COSY, HMBC and NOESY correlations of compound **1**.

**Figure 3 plants-12-01500-f003:**
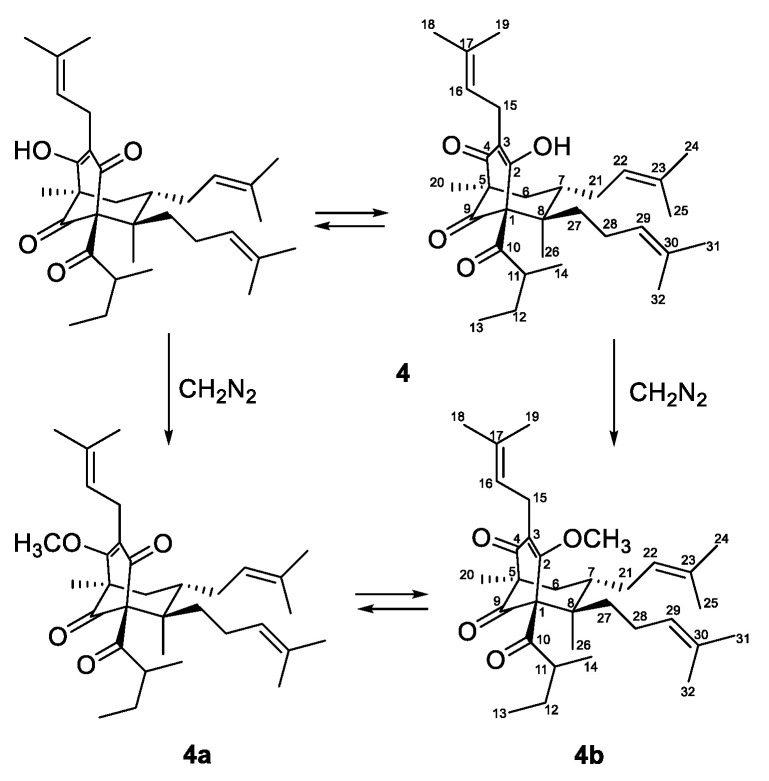
Methylation of compound **4**.

**Figure 4 plants-12-01500-f004:**
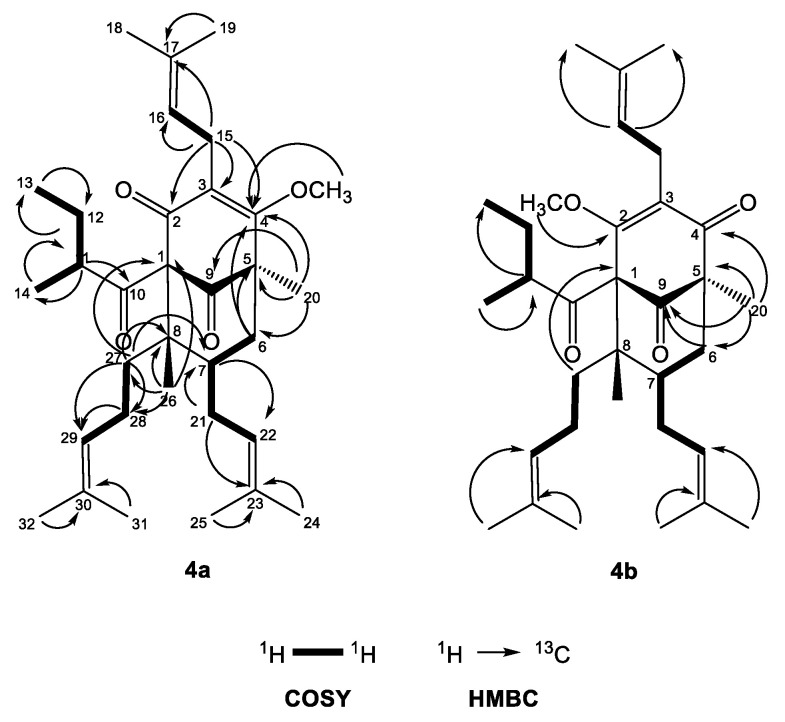
Selected COSY and HMBC correlations of **4a** and **4b**.

**Figure 5 plants-12-01500-f005:**
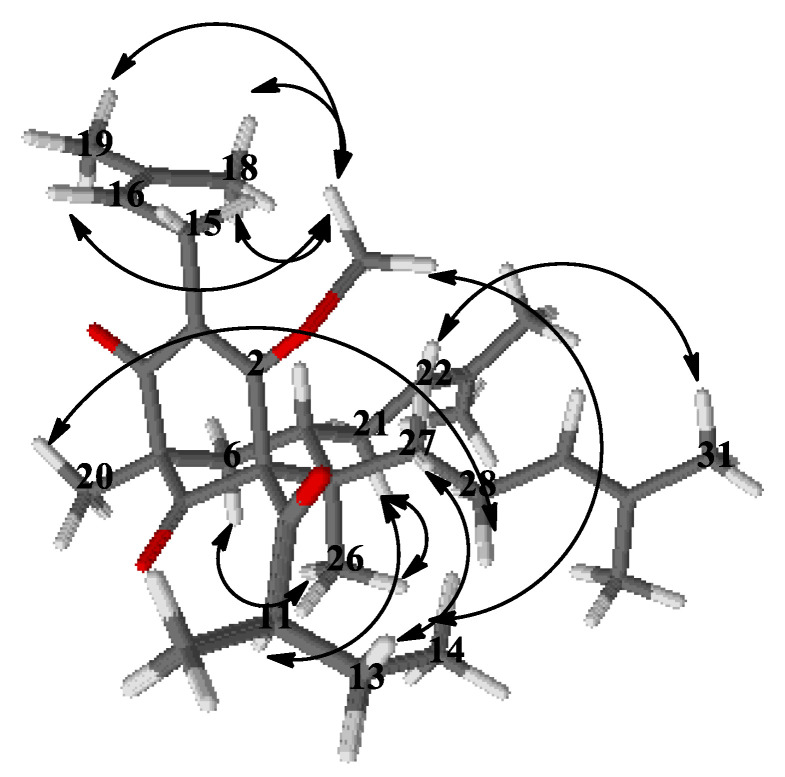
Selected NOESY correlation of **4a**.

**Figure 6 plants-12-01500-f006:**
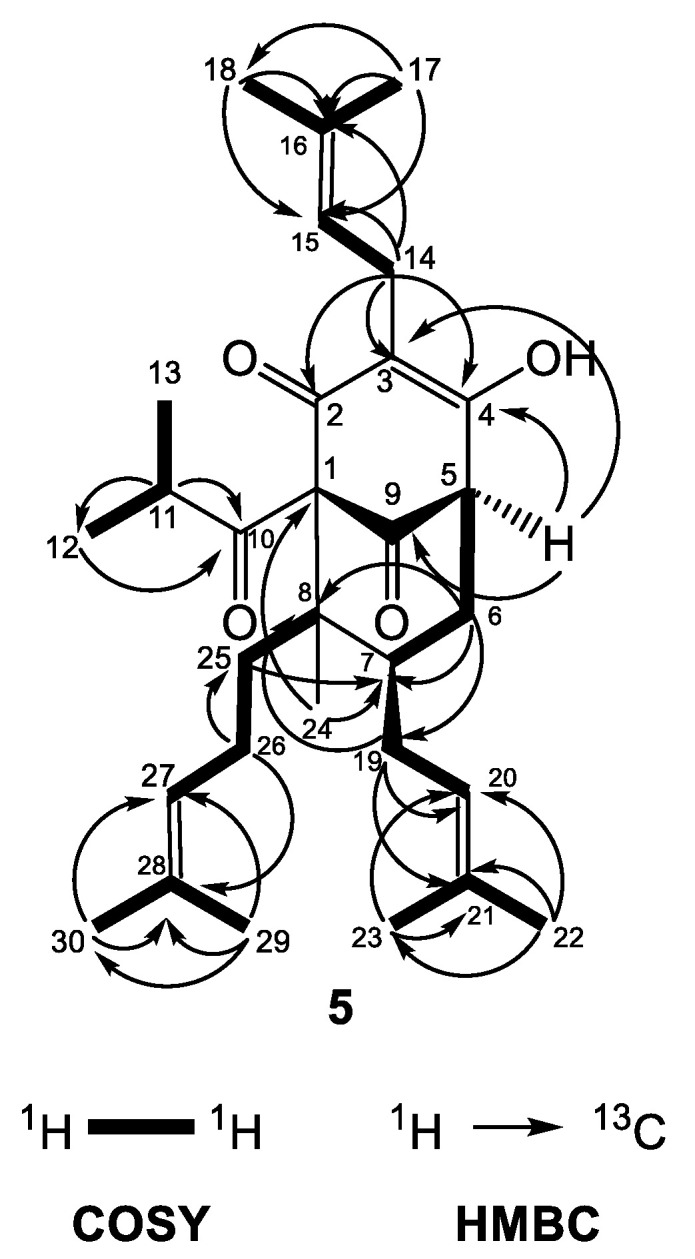
Selected COSY and HMBC correlations of **5**.

**Figure 7 plants-12-01500-f007:**
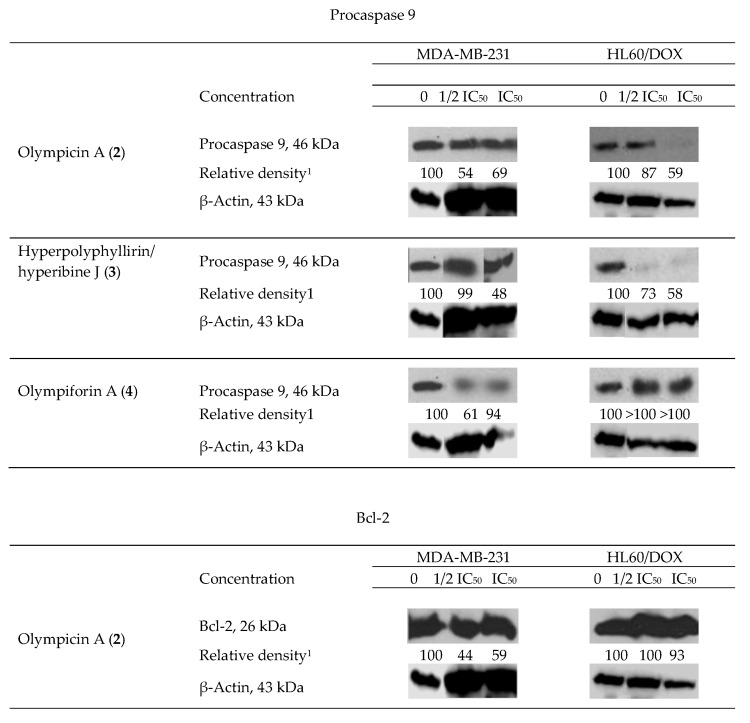
Western blot results—impact of olympicin A (**2**), hyperpolyphyllirin/hyperibine J (**3**) and olympiforin A (**4**) on the precursor caspase 9 and of **2** on Bcl-2 in tumor cell lines MDA-MB-231 and HL60/DOX. The cells were exposed to equieffective concentrations, multiples of IC_50_ (IC_50_ and ½ IC_50_, respectively). Densitometry analysis presented the quantity of the bands after comparison with the ubiquitous protein β-actin.

**Figure 8 plants-12-01500-f008:**
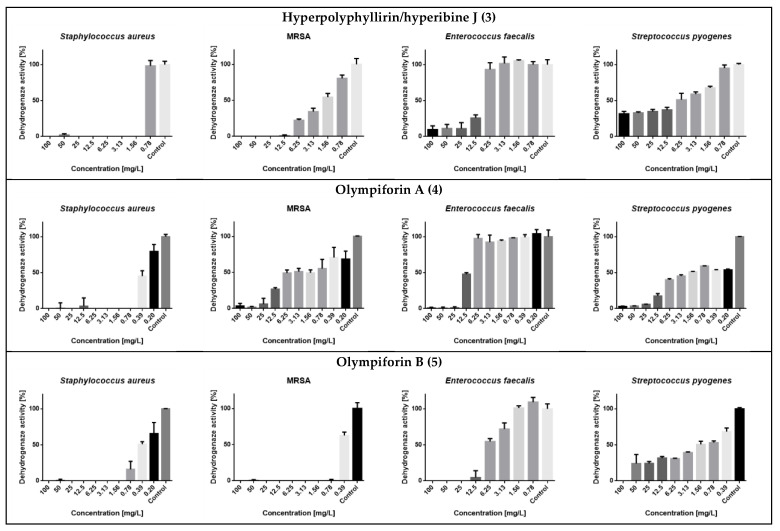
Dehydrogenase (metabolic) activity of the affected strains treated with the tested substances.

**Figure 9 plants-12-01500-f009:**
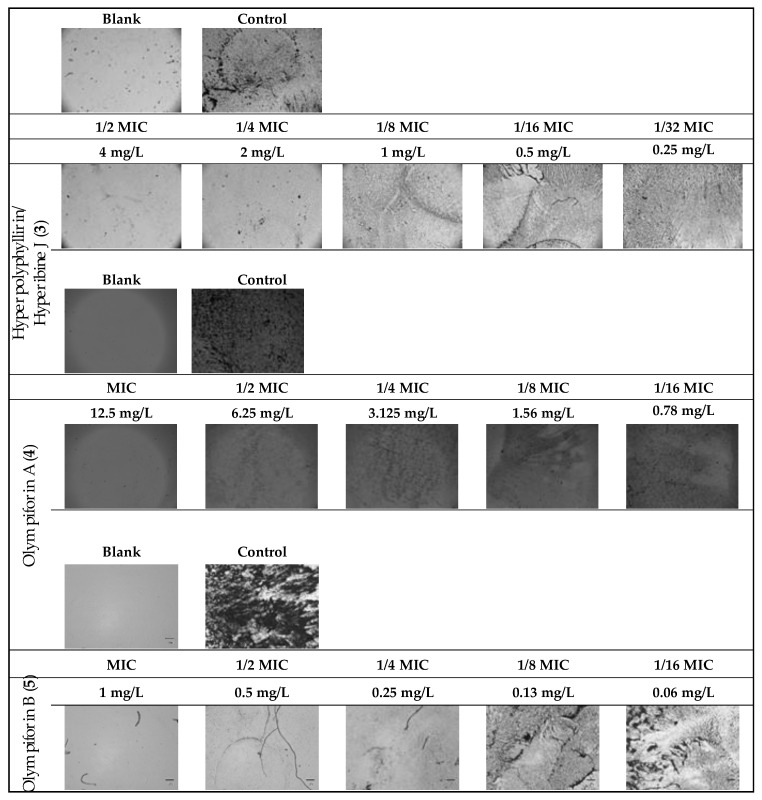
MRSA biofilm inhibition of eqieffective concentrations of the MIC for hyperpolyphyllirin/hyperibine J (**3**), olympiforin A (**4**) and olympiforin B (**5**) (two-fold serial dilutions).

**Figure 10 plants-12-01500-f010:**
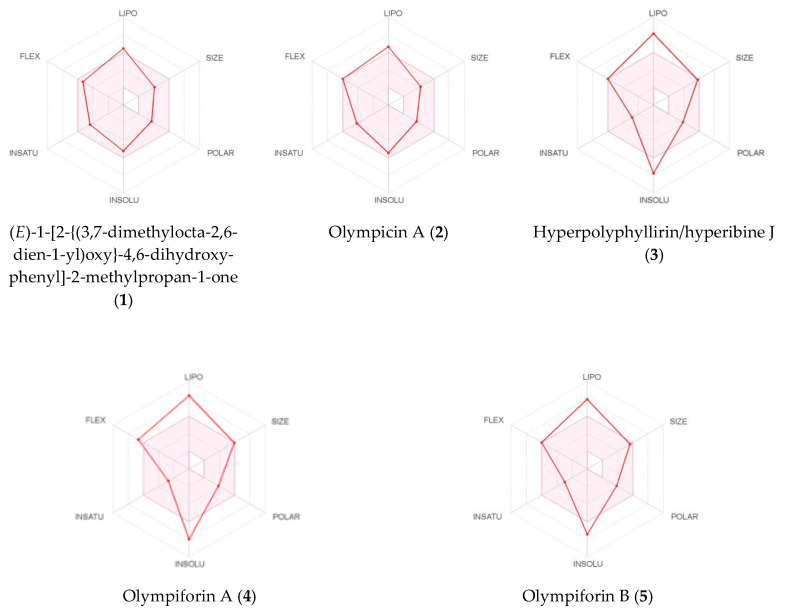
Bioavailability radars of (*E*)-1-[2-{(3,7dimethylocta-2,6-dien-1-yl)oxy}-4,6-dihydroxyphenyl]-2-methylpropan-1-one (**1**), olympicin A (**2**), hyperpolyphyllirin/hyperibine J (**3**), olympiforin A (**4**) and olympiforin B (**5**).

**Table 1 plants-12-01500-t001:** ^1^H- and ^13^C-NMR spectroscopic data of **1** recorded in CDCl_3_ (293 K, ^1^H at 600 MHz and ^13^C at 150 MHz).

No.	*δ*_C_, Mult. ^1^	*δ*_H_, (*J* in Hz)
1	105.2, C	
2	162.5, C	
3	91.7, CH	5.93 *d* (2.3)
4	162.55, C	
5	96.5, CH	6.00 *d* (2.3)
6	167.4, C	
1′	210.6, C	
2′	39.5, CH	3.80 *sept* (6.8)
3′ and 4′	19.3, CH_3_	1.15 *d* (6.8)
1″	65.6, CH_2_	4.56 *d* (6.7)
2″	118.1, CH	5.50 *m*
3″	142.3, C	
4″	39.4, CH_2_	2.10 *m*
5″	26.3, CH_2_	2.13 *m*
6″	123.6, CH	5.10 *m*
7″	132.0, C	
8″	17.7, CH_3_	1.62 *s*
9″	25.7, CH_3_	1.69 *s*
10″	16.6, CH_3_	1.74 *s*
6-OH		14.14 *br. s*

^1^ Multiplicities were determined by HSQC experiments.

**Table 2 plants-12-01500-t002:** ^13^C-NMR (150 MHz in CD_3_OD) spectral data of compounds **4**, **4a** and **4b**.

Carbon No.	*δ*_C_, Mult. ^1^		
	4	4a	4b
1	82.7 ^2^, C	84.0, C	78.5, C
2	ND ^3^	194.3, C	170.2, C
3	121.3, C	128.0, C	123.3, C
4	ND ^3^	173.2, C	196.6, C
5	ND ^3^	54.5, C	60.3, C
6	41.9, CH_2_	40.4, CH_2_	42.1, CH_2_
7	43.8, CH	43.4, CH	44.3, CH
8	48.7, C	48.6, C	60.3, C
9	209.2, C	207.8, C	208.1, C
10	211.6, C	209.0, C	209.3, C
11	49.8, CH	49.2, CH	47.4, CH
12	28.7, CH_2_	27.4, CH_2_	27.6, CH_2_
13	12.1, CH_3_	11.6, CH_3_	11.8, CH_3_
14	17.3, CH_3_	16.6, CH_3_	17.5, CH_3_
15	22.6, CH_2_	23.4, CH_2_	23.5, CH_2_
16	122.6, CH	121.6, CH	122.3 ^4^, CH
17	133.5, C	133.0, C	132.9, C
18	18.1 ^4^, CH_3_	18.0 ^4^, CH_3_	18,1, CH_3_
19	26.0, CH_3_	25.7 ^4^, CH_3_	25.7, CH_3_
20	16.9, CH_3_	16.9, CH_3_	16.5, CH_3_
21	25.8, CH_2_	27.0, CH_2_	27.8, CH_2_
22	126.0, CH	122.4, CH	122.3 ^4^, CH
23	131.9, C	133.4, C	133.4, C
24	25.9, CH_3_	25.8, CH_3_	25.8, CH_3_
25	17.9, CH_3_	18.0^4^, CH_3_	17.0, CH_3_
26	14.9, CH_3_	13.4, CH_3_	13.6, CH_3_
27	38.0, CH_2_	36.4, CH_2_	25.3, CH_2_
28	28.5, CH_2_	25.1, CH_2_	37.7, CH_2_
29	123.8, CH	124.8, CH	124.7, CH
30	134.2, C	131.2, C	131.4, C
31	18.1 ^4^, CH_3_	17.7, CH_3_	17.8, CH_3_
32	26.1, CH_3_	25.7 ^4^, CH_3_	25.9, CH_3_
OCH_3_	-	62.3, CH_3_	60.6, CH_3_

^1^ Multiplicities were determined by HSQC experiments; ^2^ despite the keto-enol tautomerism at C-2 and C-4, the signal is seen as a weak cross-peak in HMBC, correlating with *δ*_H_ 0.99 (s, H_3_-26); ^3^ not detected; ^4^ signals overlapped.

**Table 3 plants-12-01500-t003:** ^1^H-NMR (600 MHz in CD_3_OD) spectral data of compounds **4**, **4a** and **4b**.

Position	*δ*_H_, Mult. (*J* in Hz)		
	4	4a	4b
6	1.94 *dd* (13.7, 4.3, H_eq_); 1.38 *dd* (12.2, 13.7, H_ax_)	1.95 *m* (H_eq_); 1.38 *dd* (13.7, 12.7, H_ax_)	1.88 *dd* (14.0, 4.3, H_eq_); 1.31 *dd* (14.0, 12.0, H_ax_)
7	1.68 *m*	1.56 *m*	1.69 *m*
11	1.82 *m*	1.74 *m*	2.12 *m*
12	1.76 *m*; 1.25 m	1.71 *m*; 1.28 m	2.13 *m*
13	0.79 *t* (7.5)	0.78 *t* (7.4)	0.87 *t* (7.5)
14	1.08 *d* (6.5)	1.12 *d* (6.50)	1.17 *d* (6.5)
15	3.10 *m*	3.21 *dd* (15.0, 6.9); 3.11 *dd* (15.0, 6.3)	3.23 *dd* (16.4, 5.8); 3.42 *dd* (16.6, 6.3)
16	5.06 *m*	5.01 *m*	5.00 *m*
18	1.72 *s*	1.70 *s*	1.69 *s*
19	1.64 *s*	1.66 *s*	1.69 *s*
20	1.25 *s*	1.30 *s*	1.21 *s*
21	2.03 *m*; 1.95 *m*	2.12 *m*; 1.75 *m*	2.14 *m*; 1.71 *m*
22	5.02 *m*	4.95 *m*	4.93 *m*
24	1.66 *s*	1.68 *s*	1.67 *s*
25	1.60 *s*	1.54 *s*	1.56 *s*
26	0.99 *s*	1.01 *s*	1.05 *s*
27	1.53 *m*; 1.80 *m*	1.96 *m*; 1.32 *m*	1.96 *m*; 2.18 *m*
28	2.09 *m*; 1.76 *m*	2.12 *m*; 1.80 *m*	1.37 *m*
29	4.99 *m*	5.05 *m*	5.02 *m*
31	1.58 *s*	1.60 *s*	1.61 *s*
32	1.68 *s*	1.65 *s*	1.67 *s*
OCH_3_		3.92 *s*	4.03 *s*

**Table 4 plants-12-01500-t004:** ^13^C-NMR (150 MHz in CD_3_OD) and ^1^H-NMR (600 MHz in CD_3_OD) spectral data for **5**.

No.	*δ*_C_, Mult. ^1^	*δ*_H_ (*J* in Hz)
1	83.7, C	
2	192.2, C	
3	117.9, C	
4	169.2, C	
5	53.4, CH	3.24 *m* ^2^
6	36.5, CH_2_	1.96 *dd* (4.4, 14, H_eq_); 1.36 *m* (H_ax_)
7	42.5, CH	1.63 *m*
8	48.6, C	
9	206.2, C	
10	209,5, C	
11	42,2, CH	2.08 *sept* (6.5)
12	20.3, CH_3_	1.11 *d* (6.5)
13	21.4, CH_3_	1.03 *d* (6.5)
14	22.1, CH_2_	3.19 *m*, 3.24 *m* ^2^
15	120.4, CH	5.25 *m*
16	137.8, C	
17	18.0, CH_3_	1.78 *s* ^2^
18	25.9, CH_3_	1.78 *s* ^2^
19	25.1, CH_2_	1.90 *m*; 2.13 *m*
20	124.7, CH	5.06 *m*
21	131.2, C	
22	17.7, CH_3_	1.60 *s*
23	25.7, CH_3_	1.65 *s*
24	13.6, CH_3_	1.05 *s*
25	31.8, CH_2_	1.73 *m*; 2.04 *m*
26	27.2, CH_2_	1.80 *s*; 2.14 *m*
27	122.4, CH	4.96 *m*
28	133.4, C	
29	17.9, CH_3_	1.57 *s*
30	25.8, CH_3_	1.69 *s*

^1^ Multiplicities were determined by HSQC experiments. ^2^ Signals overlapped.

**Table 5 plants-12-01500-t005:** Cytotoxic activity of (*E*)-1-[2-{(3,7-dimethylocta-2,6-dien-1-yl)oxy}-4,6-dihydroxyphenyl]-2-methylpropan-1-one (**1**), olympicin A (**2**), hyperpolyphyllirin/hyperibine J (**3**), olympiforin A (**4**) and olympiforin B (**5**) on some tumor and non-cancerous cell lines after 72 h exposure (MTT test) expressed as IC_50_ (μM). Etoposide was used as a positive standard.

Compound	Cell Line						
	MDA-MB-231	EJ	K-562	HL-60	HL-60/DOX	HEK-293	EA.hy926
**1**	24.94 ± 3.2	7.01 ± 1.3	9.05 ± 0.8	16.64 ± 3.3	5.59 ± 0.4	14.74 ± 2.2	6.73 ± 0.9
**2**	2.95 ± 0.4	5.61 ± 0.5	22.87 ± 4.0	13.77 ± 1.8	1.96 ± 0.2	22.47 ± 3.8	33.97 ± 6.5
**3**	2.63 ± 0.3	8.02 ± 1.6	3.17 ± 0.3	10.90 ± 1.9	1.28 ± 0.2	10.68 ± 2.1	19.86 ± 3.1
**4**	12.51 ± 2.1	8.06 ± 1.7	17.15 ± 3.0	22.25 ± 3.4	12.19 ± 3.2	35.77 ± 4.4	5.34 ± 1.9
**5**	1.31 ± 0.2	1.21 ± 0.3	1.58 ± 0.1	1.16 ± 0.1	1.71 ± 0.2	0.89 ± 0.1	9.43 ± 2.0
Etoposide	8.42 ± 1.3	5.4 ± 1.3	2.34 ± 0.7	1.27 ± 0.4	42.34 ± 2.7	N.T. ^1^	N.T. ^1^

^1^ Not tested.

**Table 6 plants-12-01500-t006:** Selectivity indices (SI) ^1^ and resistance indices (RI) ^2^ of (*E*)-1-[2-{(3,7-dimethylocta-2,6-dien-1-yl)oxy}-4,6-dihydroxyphenyl]-2-methylpropan-1-one (**1**), olympicin A (**2**), hyperpolyphyllirin/hyperibine J (**3**), olympiforin A (**4**) and olympiforin B (**5**).

Compound	SI_HEK-293/x_	SI_EA_._hy926/x_	RI	x
**1**	1.02	0.46	0.34	14.48
**2**	1.99	3.00	0.14	11.30
**3**	1.73	3.21	0.12	6.18
**4**	2.39	0.35	0.55	14.99
**5**	0.68	7.14	1.47	1.32

^1^ SI = IC_50_(HEK-293)/IC_50_(x) or IC_50_(EA.hy926)/IC_50_(x), where x is the average IC_50_ of all chemo-sensitive tested tumor cell lines. ^2^ RI = IC_50_(HL-60/DOX)/IC_50_(HL-60).

**Table 7 plants-12-01500-t007:** Antiproliferative activity of (*E*)-1-[2-{(3,7-dimethylocta-2,6-dien-1-yl)oxy}-4,6-dihydroxyphenyl]-2-methylpropan-1-one (**1**), olympicin A (**2**), hyperpolyphyllirin/hyperibine J (**3**), olympiforin A (**4**) and olympiforin B (**5**) on some tumor cell lines expressed as GI_50_ and TGI (μM). Half growth inhibition (GI_50_) is the drug concentration resulting in a 50% reduction in the net protein increase (measured by MTT test) in control cells during the drug incubation. Total growth inhibition (TGI) is the drug concentration resulting in total growth suppression.

Compound	Cell Line					
	MDA-MB-231		K-562		HL-60	
	GI_50_	TGI	GI_50_	TGI	GI_50_	TGI
**1**	10.41	18.56	5.20	11.90	3.79	9.39
**2**	1.07	2.54	20.29	23.49	72.62	81.63
**3**	1.93	2.96	0.44	0.77	41.07	47.43
**4**	0.52	0.81	15.98	18.38	41.07	55.55

**Table 8 plants-12-01500-t008:** Minimal inhibitory concentrations (MIC) of hyperpolyphyllirin/hyperibine J (**3**), olympiforin A (**4**) and olympiforin B (**5**) against four Gram-positive bacteria.

Compound	MIC		Compound	MIC	
	[mg/L]	[μM]		[mg/L]	[μM]
	** *Staphylococcus aureus* **			** *Enterococcus faecalis* **	
**3**	2.00	4.14	**3**	8.00	16.57
**4**	0.78	1.57	**4**	12.5	25.17
**5**	1.00	2.14	**5**	4.00	8.53
**Gentamicin**	0.25	0.52	**Gentamicin**	8.00	16.75
			**Penicillin**	2.50	7.48
			**Ciprofloxacin**	0.50	1.51
	**MRSA**			** *Streptococcus pyogenes* **	
**3**	8.00	16.57	**3**	31.00	64.22
**4**	12.5	25.17	**4**	12.50	25.17
**5**	1.00	2.14	**5**	4.00	8.53
**Gentamicin**	0.25	0.52	**Penicillin**	0.08	0.24

Bold refers the bacterial strains and the antibiotics.

**Table 9 plants-12-01500-t009:** Biofilm inhibition activity of hyperpolyphyllirin/hyperibine J (**3**), olympiforin A (**4**) and olympiforin B (**5**) on MRSA.

Compound	MBIC ^1^		MBIC_50_ ^2^	
	[mg/L]	[μM]	[mg/L]	[μM]
**3**	4.0 (= ½ MIC)	8.29	0.17 (0.07 to 0.46)	0.35 (0.14 to 0.95)
**4**	12.5 (= MIC)	25.17	0.67 (0.39 to 1.17)	1.35 (0.78 to 2.36)
**5**	0.5 (= ½ MIC)	1.07	0.05 (0.03 to 0.07)	0.12 (0.09 to 0.17)

^1^ Minimum biofilm inhibitory concentration. ^2^ Median biofilm inhibitory concentration.

**Table 10 plants-12-01500-t010:** Basic physicochemical properties and computational descriptors of (*E*)-1-[2-{(3,7dimethylocta-2,6-dien-1-yl)oxy}-4,6-dihydroxyphenyl]-2-methylpropan-1-one (**1**), olympicin A (**2**), hyperpolyphyllirin/hyperibine J (**3**), olympiforin A (**4**) and olympiforin B (**5**).

Properties	1	2	3	4	5
Molecular weight (g/mol)	332.43	346.46	482.69	496.72	468.67
No. heavy atoms	24	25	35	36	34
No. aromatic heavy atoms	6	6	0	0	0
Fraction Csp3	0.45	0.48	0.65	0.66	0.63
No. rotatable bonds	8	9	9	10	9
No. H-bond acceptors	4	4	4	4	4
No. H-bond donors	2	2	1	1	1
Molar Refractivity	99.10	103.91	146.40	151.21	141.85
Topological polar surface area (Å²)	66.76	66.76	71.44	71.44	71.44

**Table 11 plants-12-01500-t011:** Lipophilicity of (*E*)-1-[2-{(3,7dimethylocta-2,6-dien-1-yl)oxy}-4,6-dihydroxyphenyl]-2-methylpropan-1-one (**1**), olympicin A (**2**), hyperpolyphyllirin/hyperibine J (**3**), olympiforin A (**4**) and olympiforin B (**5**).

Properties	1	2	3a ^1^	3b ^1^	4a ^2^	4b ^2^	5
Log Po/w (iLOGP)	3.24	3.58	5.11	4.71	4.53	4.06	4.55
Log Po/w (XLOGP3)	5.82	6.17	8.75	8.20	9.11	8.56	8.39
Log Po/w (WLOGP)	5.01	5.40	7.65	7.65	8.04	8.04	7.26
Log Po/w (MLOGP)	2.93	3.16	3.91	3.91	4.10	4.10	3.72
Log Po/w (SILICOS-IT)	4.75	5.17	8.24	8.24	8.67	8.67	7.70
Consensus Log Po/w	4.35	4.69	6.73	6.54	6.89	6.69	6.33

^1^ Tautomeric forms of compound **3**. ^2^ Tautomeric forms of compound **4**.

**Table 12 plants-12-01500-t012:** Water solubility prediction values of (*E*)-1-[2-{(3,7dimethylocta-2,6-dien-1-yl)oxy}-4,6-dihydroxyphenyl]-2-methylpropan-1-one (**1**), olympicin A (**2**), hyperpolyphyllirin/hyperibine J (**3**), olympiforin A (**4**) and olympiforin B (**5**) based on three alternative models.

Properties	1	2	3a ^1^	3b ^1^	4a ^2^	4b ^2^	5
Log S (ESOL)	−5.22	−5.46	−7.75	−7.40	−8.00	−7.65	−7.44
Solubility	1.98 × 10^−3^ mg/mL; 5.96 × 10^−6^ mol/L	1.20 × 10^−3^ mg/mL; 3.48 × 10^−6^ mol/L	8.56 × 10^−6^ mg/mL; 1.77 × 10^−8^ mol/L	1.90 × 10^−5^ mg/mL; 3.94 × 10^−8^ mol/L	4.98 × 10^−6^ mg/mL; 1.00 × 10^−8^ mol/L	1.11 × 10^−5^ mg/mL; 2.23 × 10^−8^ mol/L	1.71 × 10^−5^ mg/mL; 3.65 × 10^−8^ mol/L
Class	Moderately soluble	Moderately soluble	Poorly soluble	Poorly soluble	Poorly soluble	Poorly soluble	Poorly soluble
Log S (Ali)	−6.99	−7.36	−10.13	−9.56	−10.50	−9.93	−9.76
Solubility	3.38 × 10^−5^ mg/mL; 1.02 × 10^−7^ mol/L	1.53 × 10^−5^ mg/mL; 4.41 × 10^−8^ mol/L	3.57 × 10^−8^ mg/mL; 7.39 × 10^−11^ mol/L	1.33 × 10^−7^ mg/mL; 2.75 × 10^−10^ mol/L	1.55 × 10^−8^ mg/mL; 3.13 × 10^−11^ mol/L	5.78 × 10^−8^ mg/mL; 1.16 × 10^−10^ mol/L	8.19 × 10^−8^ mg/mL; 1.75 × 10^−10^ mol/L
Class	Poorly soluble	Poorly soluble	Insoluble	Poorly soluble	Insoluble	Poorly soluble	Poorly soluble
Log S (SILICOS-IT)	−4.29	−4.68	−6.88	−6.88	−7.27	−7.27	−6.29
Solubility	1.72 × 10^−2^ mg/mL; 5.16 × 10^−7^ mol/L	7.19 × 10^−3^ mg/mL; 2.07 × 10^−5^ mol/L	6.29 × 10^−5^ mg/mL; 1.30 × 10^−7^ mol/L	6.29 × 10^−5^ mg/mL; 1.30 × 10^−7^ mol/L	2.64 × 10^−5^ mg/mL; 5.32 × 10^−8^ mol/L	2.64 × 10^−5^ mg/mL; 5.32 × 10^−8^ mol/L	2.38 × 10^−4^ mg/mL; 5.08 × 10^−7^ mol/L
Class	Moderately soluble	Moderately soluble	Poorly soluble	Poorly soluble	Poorly soluble	Poorly soluble	Poorly soluble

^1^ Tautomeric forms of **3**. ^2^ Tautomeric forms of **4**.

**Table 13 plants-12-01500-t013:** Calculated ADME and pharmacokinetic parameters of (*E*)-1-[2-{(3,7dimethylocta-2,6-dien-1-yl)oxy}-4,6-dihydroxyphenyl]-2-methylpropan-1-one (**1**), olympicin A (**2**), hyperpolyphyllirin/hyperibine J (**3**), olympiforin A (**4**) and olympiforin B (**5**).

Properties	1	2	3a	4a	5
GI absorption	High	High	Low	Low	Low
BBB permeant	Yes	No	No	No	No
P-gp substrate	No	Yes	Yes	Yes	Yes
CYP1A2 inhibitor	Yes	Yes	No	No	No
CYP2C19 inhibitor	No	No	No	No	No
CYP2C9 inhibitor	Yes	Yes	No	No	Yes
CYP2D6 inhibitor	No	No	No	No	No
CYP3A4 inhibitor	Yes	Yes	Yes	Yes	Yes
Log *K*_p_ (skin permeation)	−4.20 cm/s	−4.03 cm/s	−3.03 cm/s ^1^	−2.86 cm/s ^2^	−3.20 cm/s

^1^ The other tautomeric form **3b** has Log *K*_p_ = −3.42 cm/s. ^2^ The other tautomeric form **4b** has Log *K*_p_ = −3.25 cm/s.

**Table 14 plants-12-01500-t014:** Drug-likeness, medicinal chemistry and lead-likeness parameters of (*E*)-1-[2-{(3,7dimethylocta-2,6-dien-1-yl)oxy}-4,6-dihydroxyphenyl]-2-methylpropan-1-one (**1**), olympicin A (**2**), hyperpolyphyllirin/hyperibine J (**3**), olympiforin A (**4**) and olympiforin B (**5**).

Properties	1	2	3a	4a	5
Lipinski	Yes; 0 violations	Yes; 0 violations	Yes; 0 violations	Yes; 0 violations	Yes; 0 violations
Ghose	Yes	Yes	No; 4 violations: MW > 480, WLOGP > 5.6, MR > 130, No. atoms > 70	No; 4 violations: MW > 480, WLOGP > 5.6, MR > 130, No. atoms > 70	No; 3 violations: WLOGP > 5.6, MR > 130, No. atoms > 70
Veber	Yes	Yes	Yes	Yes	Yes
Egan	Yes	Yes	No; 1 violation: WLOGP > 5.88	No; 1 violation: WLOGP > 5.88	No; 1 violation: WLOGP > 5.88
Muegge	No; 1 violation: XLOGP3 > 5	No; 1 violation: XLOGP3 > 5	No; 1 violation: XLOGP3 > 5	No; 1 violation: XLOGP3 > 5	No; 1 violation: XLOGP3 > 5
Bioavailability Score	0.55	0.55	0.56	0.56	0.56
PAINS	0 alert	0 alert	0 alert	0 alert	0 alert
Brenk	1 alert: isolated alkene	1 alert: isolated alkene	2 alerts: β- keto anhydride, isolated alkene	2 alerts: β-keto anhydride, isolated alkene	2 alerts: β-keto anhydride, isolated alkene
Leadlikeness	No; 2 violations: Rotors ^1^ > 7, XLOGP3 > 3.5	No; 2 violations: Rotors ^1^ > 7, XLOGP3 > 3.5	No; 3 violations: MW > 350, Rotors ^1^ > 7, XLOGP3 > 3.5	No; 3 violations: MW > 350, Rotors ^1^ > 7, XLOGP3 > 3.5	No; 3 violations: MW > 350, Rotors ^1^ > 7, XLOGP3 > 3.5
Synthetic Accessibility	3.11	3.70	6.86	7.10	6.74

^1^ Rotors—number of rotatable bonds.

## Data Availability

All raw data from the experiments are available from the authors.

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
