# Peer review of "Cytotoxic and Antibacterial Prenylated Acylphloroglucinols from Hypericum olympicum L."

_plants, 2023, doi:10.3390/plants12071500_

Round 1

Reviewer 1 Report

Dear Editors,

The work entitled “Cytotoxic and antibacterial prenylated acylphloroglucinols from Hypericum olympicum L.“ (plants-2313642) is very interesting and innovative.

Due to the still-increasing resistance to commercially used antibiotics among bacteria, especially those living in biofilm consortia, it is important to search for new antibacterial compounds.

All compounds used in the current study were studied for their cytotoxic activity. Molecules (3) hyperpolyphyllirin/hyperibine J, (4) olympiforin A, and (5) olympiforin B were evaluated for their antibacterial (anti-biofilm) activities.

Moreover, the structures of the isolated compounds were also evaluated in silico using the web tool SwissADME to check their bioavailability, which significantly increases the value of the work.

It seems to me that the work fits in the scope of the Plants journal and may be considered for publication after the introduction of the corrections suggested below:

1.    Figure 5 – Caption - either we use compound names or numbers, or both, but then numbers 2, 3, and 4 should be in the Figure, not just in the caption. Decide, please.

2.    Figure 6 – Were any of the results statistically significant? If so, it should be shown in the Figure with the use of e.g. asterisk

3.    Table 9 contains the values of MBIC and MBIC50 of investigated substances. It does not equal the inhibition of biofilm. The authors state that the antibiofilm activity of 3-5 substances was conducted according to the method by Stepanovic et al (2007). However, there are no results of the optical density (OD) measurement of each sample stained with crystal violet, which should be measured at 570 nm using a microtiter plate reader. Please introduce the results to the work.

4.    Caption of Figure 9 – explain 3, 4, and 5 by adding the names of the compounds, please.

5.    Figure 7. Incorrect caption - correct it. Explain 3, 4, and 5 by adding the names of the compounds, please.

6.    Apply the explanation of plant compound numbers to Tables 10-14 as well.

7.    The antibiofilm properties of 3-5 compounds should be described in the text of the manuscript

8.    Line 356 - How the significance of the antibiofilm activity of the 3-5 substances was assessed?

9.    Line 555 - S. aureus – in italics

10. Lines 853, 855 – use the capital letter “L” in the unit μL

11. Lines 862-864 – “After that, the cells were air dried and the biofilm formation in the wells was documented microscopically (40×)” - What tool and computer program were used to visualize the bacterial biofilms?

12. Line 865 - authors wrote: "… the absorbance of each well was measured at 550 nm with lid.” - Where are the values of the absorbance? They could show the antibiofilm activities of analyzed plant-derived substances…

13. Lines 30 and 894 – the unit “mg/mL” was used. In all other places in the text “mg/L” has been used; it does not correspond to data shown in Table 9 - please respond to it or correct…

Best regards,

Author Response

We are grateful to the reviewer very much for pointing out some technical errors which would have led to wrong conclusions about the results. In addition, we are grateful for the other suggestions for improving the manuscript.

  1. Figure 5 – Caption - either we use compound names or numbers, or both, but then numbers 2, 3, and 4 should be in the Figure, not just in the caption. Decide, please.

Answer: The numbers were included along with the compound names in Figure 5.

  1. Figure 6 – Were any of the results statistically significant? If so, it should be shown in the Figure with the use of e.g. asterisk

Answer: The statistical analysis was performed using One-way ANOVA. Because of the great significant difference between the treated samples and the control and the difficulty to include all data asterisks in the figure we decided to include these in Table S2 placed in Supplementary Materials.

  1. Table 9 contains the values of MBIC and MBIC50 of investigated substances. It does not equal the inhibition of biofilm. The authors state that the antibiofilm activity of 3-5 substances was conducted according to the method by Stepanovic et al (2007). However, there are no results of the optical density (OD) measurement of each sample stained with crystal violet, which should be measured at 570 nm using a microtiter plate reader. Please introduce the results to the work.

Answer: We are not sure what is exactly the meaning of “the table with the values of MBIC and MBIC50 does not equal the inhibition of biofilm”. As we have written in Materials and methods “The minimum biofilm inhibition concentration MBIC is the lowest concentration of an antimicrobial agent that results in no detectable biofilm growth” i.e. MBIC inhibits the biofilm—it is the lowest concentration that inhibits the biofilm completely.

As to the optical density—we now added it in Table S1 placed into the Supplementary Materials. Some authors give it in their articles and some do not, and we preferred to be supplementary in order not to make the manuscript too heavy with raw and voluminous data but to emphasize the conclusions.

  1. The caption of Figure 9 – explain 3, 4, and 5 by adding the names of the compounds, please.

Answer: This is explained.

  1. Figure 7. Incorrect caption - correct it. Explain 3, 4, and 5 by adding the names of the compounds, please.

Answer: We apologize for that mistake and corrected it.

  1. Apply the explanation of plant compound numbers to Tables 10-14 as well.

Answer: We applied this to all the tables for cytotoxicity.

  1. The antibiofilm properties of 3-5 compounds should be described in the text of the manuscript

Answer: Yes, we described it and that improved the presentation of results.

  1. Line 356 - How the significance of the antibiofilm activity of the 3-5 substances was assessed?

Answer: That is connected to the previous question. You are right that these conclusions and explaining how we reached them—based on comparison with the activity of other published compounds, as well as whether the MBIC is lower than the respective MIC, is suitable only for the discussion. Hence, we removed these conclusions in the results section.

  1. Line 555 - S. aureus – in italics

Answer: The typographic error was corrected.

  1. Lines 853, 855 – use the capital letter “L” in the unit μL

Answer: The typographic error was corrected.

  1. Lines 862-864 – “After that, the cells were air dried and the biofilm formation in the wells was documented microscopically (40×)” - What tool and computer program were used to visualize the bacterial biofilms?

Answer: We added that information to the manuscript.

  1. Line 865 - authors wrote: "… the absorbance of each well was measured at 550 nm with lid.” - Where are the values of the absorbance? They could show the antibiofilm activities of analyzed plant-derived substances…

Answer: We provided the absorbance values (Table S1) in the Supplementary Materials.

  1. Lines 30 and 894 – the unit “mg/mL” was used. In all other places in the text “mg/L” has been used; it does not correspond to data shown in Table 9 - please respond to it or correct…

Answer: It is a technical error. It should be mg/L and we corrected it.

Reviewer 2 Report

Ilieva et al. isolated five rare or new acylphloroglucinols from the aerial parts of Hypericum olympicum and performed a thorough structure elucidation of these compounds. Several biological activities of these compounds were evaluated comprising antibacterial, cytotoxic and antiproliferative effects. In addition in silico studies on ADME and of physical parameters for a possible development of these compounds for therapeutic purposes were performed.

The whole article is written in a very fluid style without grammar or spelling mistakes. The manuscript delivers a huge and impressing amount of information and results related to phytochemistry and pharmacology of the title plant. The findings are of great interest for the international readership of the journal Plants. The results show that Hypericum species and compounds isolated from this species will be of great therapeutic value in the future.

I have no comments to add and/ or proposals for changes in the manuscript. Again this is an excellent study!

Kind regards

Author Response

We are extremely grateful for the flattering and friendly review.